# Beyond the Rational Illusion: Behaviorally Realistic Strategic Classification

## Abstract

Strategic classification (SC) studies the interaction between decision models and agents who strategically manipulate their features for favorable outcomes. Existing SC frameworks typically rely on the idealized assumption that agents are strictly rational. However, evidence from behavioral economics and psychology consistently shows that real-world decision-making is often shaped by cognitive biases, deviating from pure rationality. To overcome the limitation of rational SC paradigms, we propose the **behaviorally realistic strategic classification problem**, where the strategic manipulations simultaneously exhibit both rational consideration and non-rational behaviors arising from psychological biases. To address this problem, we re-frame the paradigm of strategic classification based on prospect theory into a more realistic approach named **Prospect-Guided Strategic Framework** *(Pro-SF)*. Specifically, to capture agents' actual manipulation in practice, our framework reformulates the Stackelberg-style interaction between agents and the decision-maker in SC with three emerged mechanisms, including the asymmetry between benefits and costs, different subjective reference points, and non-rational probability distortion. Experiments on synthetic and real-world datasets establish Pro-SF as a behaviorally grounded approach to strategic classification, bridging machine learning and behavioral economics for more reliable deployment in practice.

## 1 Introduction

Machine learning models are playing an increasingly critical role in diverse human-serving domains, such as hiring Sánchez-Monedero et al. (2020), credit scoring Jagtiani & Lemieux (2019), and college admissions Kuvcak et al. (2018). In such settings, individuals may strategically modify their observable features to obtain favorable decisions. This phenomenon reflects Goodhart's Law Strathern (1997): "*When a measure becomes a target, it ceases to be a good measure.*" Once a model's decision rule becomes known or anticipated, agents often engage in 'gaming' behaviors to manipulate outcomes. For example, a loan applicant might temporarily inflate their reported income to appear more creditworthy. To anticipate such manipulations, strategic classification (SC) Hardt et al. (2016) provides a specialized machine learning (ML) framework by considering the Stackelberg-style interaction between decision makers and strategic agents Ghalme et al. (2021); Singh & Kulkarni (2024); Chen et al. (2020), serving as a key bridge between ML and social sciences.

Given its growing influence, it is increasingly essential to critically examine the foundational principles of SC. To be specific, the framework of SC rests on a simplified assumption that *agents are perfectly rational* Hardt et al. (2016); Milli et al. (2019). Consequently, under this view, an agent modifies features if, and only if, the expected benefit outweighs the cost. However, like a double-edged sword, the assumption of fully rational agents might result in conflicts with realistic scenarios, as individuals often behave in ways that deviate sharply from rational utilities:

- **Example 1.** In financial investment Banerji et al. (2020), individuals often react more strongly to a potential $80 loss than to a potential $100 gain of equal probability. The same magnitude of outcome thus triggers disproportionate behavioral responses.
- **Example 2.** In credit scoring Banerji et al. (2020), consider loan approval requires applicants to exceed a threshold $A$. Those whose subjective reference point $B$ is just below $A$ tend to make marginal adjustments, whereas those far below the threshold usually forgo effort.

(a) Financial Investment ( Loss Aversion)    (b) Credit Scoring (Reference Bias)    (c) Disease Screening (Probability Distortion)

Figure 1: Illustrative real-life scenarios of behavioral biases: (a) financial investment shaped by loss aversion, (b) credit scoring influenced by reference bias, and (c) disease screening affected by probability distortion.

- **Example 3.** In disease screening Dwyer et al. (2022), consider a rare disease with a true prevalence of only 0.5%. Although the objective probability of infection is negligible, some individuals persist in seeking repeated testing, subjectively inflating the small chance of illness.

Unfortunately, the non-rational behaviors characterized in the three examples, i.e., asymmetry between the benefit and cost, behaviors based on different reference points, and distorted probability towards events, are not considered in conventional SC frameworks. More broadly, extensive evidence from behavioral economics and psychology shows that real-world decision making systematically deviates from rational optimization Carroll & Johnson (1990); Tversky & Kahneman (1992); Jones (1999); Ariely & Jones (2008). Therefore, we highlight that the rational-agent assumption, while mathematically convenient, often collapses in real-world settings. This mismatch motivates the need for a refined SC framework to capture such realistic behavioral deviations more precisely, which raises a central research challenge:

> *Can strategic classification account for manipulations systematically shaped by psychological biases, moving beyond the conventional paradigms with rational utilities?*

To bridge this gap, we introduce the problem of **Behaviorally Realistic Strategic Classification** *(BR-SC)*, where strategic manipulations of agents are guided by realistic behavioral patterns rather than idealized rational rules. Accordingly, the classifier is designed to account for these manipulations, ensuring robustness in deployment. Fortunately, prospect theory Kahneman & Tversky (2013) and followed behavioral economics highlight three pervasive mechanisms for realistic behaviors: (1) *Loss aversion*. Individuals tend to weigh potential losses more heavily than equivalent gains (see Fig. 1(a)). (2) *Reference bias*. Individual decision depends on subjective reference points, which differ across individuals based on their personal circumstances, expectations, or prior experiences (Fig. 1(b)). (3) *Probability distortion*. Individuals overweight small-probability events, showing a tendency to "gamble" on unlikely opportunities (see Fig. 1(c)).

These behavioral mechanisms result in two consequential failure modes for classical strategic classification, *over-defense* and *under-defense*, which degrade robustness of decision models for SC (see detailed derivation in Sec. 3). Therefore, we propose the **Prospect-Guided Strategic Framework** *(Pro-SF)* to address the BR-SC problem. Pro-SF introduces a paradigm shift for SC by integrating three key principles (i.e., asymmetry between gains and losses, subjective reference points, and probability distortion) into the Stackelberg game framework of SC. This paradigm shift captures how agents actually manipulate their features in practice and redefines both the dynamics of agents' strategic manipulation and the decision maker's optimization objective.

**Our main contributions are summarized as follows:**

- We reveal two failure modes in classical SC, *over-defense* and *under-defense*, arising from the rational-agent assumption. Motivated by this analysis, we formalize the *Behaviorally Realistic Strategic Classification* problem.

- We propose the *Prospect-Guided Strategic Framework*, which integrates loss aversion, reference bias, and probability distortion into the Stackelberg game framework, providing a more realistic solution for SC.

- We conduct extensive experiments on synthetic and real-world datasets to demonstrate that Pro-SF achieves robust performance across diverse behavioral regimes. Ablation studies and sensitivity analyses further illustrate how each behavioral component contributes to robustness.

## 2 PRELIMINARY

We briefly review the strategic classification (SC) paradigm. Random variables are denoted by uppercase letters (e.g., $X$, $Y$), their realizations by lowercase (e.g., $x$, $y$), and boldface for vectors or matrices (e.g., $\mathbf{x}$, $\mathbf{X}$).

### 2.1 RATIONAL STRATEGIC CLASSIFICATION MODEL

The strategic classification problem is modeled as a Stackelberg game Li & Sethi (2017), where a **decision maker** defines a classification function $f : \mathbb{R}^d \to \{0, 1\}$, and **decision subjects** (agents) strategically manipulate their features from $\mathbf{x}$ to $\mathbf{x}'$ at a cost $c : \mathcal{X} \times \mathcal{X} \to \mathbb{R}_{\geq 0}$ Hardt et al. (2016); Miller et al. (2020).

The optimal manipulated feature $\mathbf{x}'$ is determined by the best-response function $b_R(\mathbf{x})$:

**Definition 2.1** (Rational Strategic Manipulation). *The optimal modified feature vector $\mathbf{x}'$ is determined by:*

$$\mathbf{x}' = b_R(\mathbf{x}) = \arg\max_{\mathbf{x}' \in \mathcal{X}} \left[ f(x') - \lambda c(x, x') \right], \tag{1}$$

*where $f(x') \in \{0, 1\}$ is the classification result after modification, $c(x, x')$ is the manipulation cost, $\lambda > 0$ is a trade-off parameter, and $\mathcal{D}$ is the feature space. Usually, the cost is modeled as the Mahalanobis Distance Gavish et al. (2021); Chen et al. (2023).*

From the decision maker's perspective, the classification rule $f$ is designed to remain robust under such strategic manipulation:

**Definition 2.2** (Decision Optimization). *To mitigate manipulation, the decision maker optimizes $f$ to maximize expected accuracy against strategic manipulation:*

$$f^* \in \arg\max_{f \in \mathcal{F}} \mathbb{E}_{(\mathbf{x}, y) \sim \mathcal{D}} \left[ \mathbb{1}\left( f(b(\mathbf{x})) = y \right) \right], \tag{2}$$

*where $\mathcal{F}$ is the set of all feasible classification rules, $\mathbb{1}$ denotes the indicator function, and $y$ is the observed label.*

### 2.2 THE ACHILLES' HEEL OF STRATEGIC CLASSIFICATION

The rational-agent assumption underlying Eq. equation 1 and Eq. equation 2 is mathematically convenient but behaviorally restrictive. A large body of work in behavioral economics and psychology has shown that human decision-making systematically departs from perfect rationality Tversky & Kahneman (1992); Ariely & Jones (2008); Kahneman & Tversky (2013); Borkar & Chandak (2021). Among these, **Prospect Theory** Tversky & Kahneman (1992); Kahneman & Tversky (2013) provides a unifying framework, capturing how individuals evaluate uncertain outcomes. Therefore, we emphasize three particularly relevant deviations from rationality that will serve as the foundation for our behaviorally realistic formulation:

- **Loss aversion in manipulation.** Agents subjectively inflate perceived costs relative to gains. This asymmetry causes them to forgo objectively beneficial manipulations because the perceived pain of the effort outweighs the comparable objective benefit.
- **Reference bias.** Agents evaluate outcomes relative to a subjective reference point $r$ (e.g., their expected outcome), not in absolute terms. This means the utility of a successful manipulation is contextual and varies across individuals, breaking the universal utility maximization principle.
- **Probability distortion.** Agents tend to overweight small probabilities, leading them to overestimate the impact of minor manipulations. As a result, they may make sub-optimal adjustments that seem sufficient to cross the decision threshold but ultimately fail, because their perceived decision weight $w$) does not match reality.

## 3 ANALYSIS ON LIMITATIONS OF RATIONAL ASSUMPTIONS

Without the rational assumption, classifiers optimized for rational settings suffer from performance degradation. We formalize two characteristic failure modes: *over-defense* and *under-defense*.

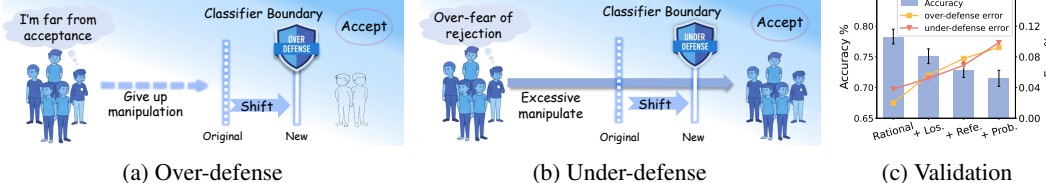

(a) Over-defense      (b) Under-defense      (c) Validation

Figure 2: Illustration of failure modes in rational strategic classification: (a) over-defense caused by agents giving up manipulation, (b) under-defense caused by excessive manipulation, and (c) validation showing how behavioral factors affect accuracy and error rates (Los. = loss aversion, Refe. = reference bias, Prob. = probability distortion).

## 3.1 OVER-DEFENSE

In practice, agents hold different subjective reference points. Some agents perceive themselves as far from acceptance, and loss aversion amplifies the effort required to close this gap. As a result, they choose not to manipulate their features, leading to *over-defense* for the classifier (shown in Fig. 2a).

**Definition 3.1** (Over-defense). *Let $f_R$ denote the classifier trained under the rational-agent assumption. We say that $f_R$ exhibits* over-defense *if it guards against the strategic manipulations that rarely occur, thereby reducing classification accuracy.*

Intuitively, over-defense shifts the decision boundary to counter manipulations that never occur. This shift flips some initial positive agents who would be accepted without any manipulation.

**Example 3.1** (Negative impact of over-defense: linear classifier). *Consider a linear classifier $f(x) = sign(w^\top x - \tau)$, a standard model in SC Hardt et al. (2016); Miller et al. (2020). Under rational assumptions, agents are expected to shift to*

$$b_R(x) = x + \frac{\max\{0, \tau - w^\top x\}}{\|w\|^2} w. \tag{3}$$

*In practice, some agents choose not to manipulate and remain at $x$. Training on $b_R(x)$ causes the classifier to shift its decision boundary outward from $\tau$ to $\tau' > \tau$, defending against manipulations that never occur. This over-defense introduces false negatives:*

$$w^\top x \geq \tau \quad but \quad w^\top x < \tau' \ \Rightarrow \ f_R'(x) = -1. \tag{4}$$

Therefore, we present the following proposition, along with a detailed proof included in Appendix C.

**Proposition 1** (Accuracy Degradation from Over-defense). *Under over-defense, a classifier trained with the rational-agent assumption achieves lower accuracy, i.e.,*

$$\mathbb{E}[\mathcal{L}(f_R(x), y)] \ > \ \mathbb{E}[\mathcal{L}(f_R(b_R(x)), y)], \tag{5}$$

*where $\mathcal{L}$ is the loss function for classification.*

## 3.2 UNDER-DEFENSE

When agents attempt manipulation, probability distortion makes them misjudge the true chance of acceptance, and loss aversion makes them overly fearful of rejection. As a result, they tend to make excessive adjustments beyond the rational manipulation, moving past the boundary anticipated by the classifier and giving rise to *under-defense* (shown in Fig. 2b).

**Definition 3.2** (Under-defense). *The classifier $f_R$ exhibits* under-defense *when it guards only against rational manipulations, but agents make larger adjustments that bypass this defense, further leading to degraded classification performance.*

**Example 3.2** (Negative Impact of Under-defense). *Consider again the linear classifier $f(x) = sign(w^\top x - \tau)$. Under rational assumptions, an agent facing $x$ is expected to move minimally to the rational endpoint $f_R(b_R(x)) = y$.*

*In practice, however, some agents adjust beyond this point to $x^*$ in the same direction. Since $f_R$ is only trained to defend manipulations up to $b_R(x)$, it leaves $x^*$ unprotected:*

$$f_R(b_R(x)) = y \quad but \quad f_R(x^*) \neq y. \tag{6}$$

*Thus, under-defense reduces accuracy by failing to guard against manipulations that exceed the rational endpoint.*

Therefore, we have the following proposition, with specific proof included in Appendix D.

**Proposition 2** (Accuracy Degradation from Under-defense)**.** *Within under-defense, a classifier trained with the rational-agent assumption fails to defend against the actual strategic manipulations. Consequently, its classification accuracy decreases:*

$$\mathbb{E}\big[\mathcal{L}(f_R(x^*), y)\big] \; > \; \mathbb{E}\big[\mathcal{L}(f_R(b_R(x)), y)\big], \tag{7}$$

*where $b_R(x)$ is the rational endpoint and $x'$ is the actual manipulated feature.*

Shown in Fig. 2c, as various behavioral mechanisms of agents are incorporated, the accuracy of the rational-based classifier gradually decreases, while both over-defense and under-defense errors increase.

## 4 PROSPECT-GUIDED STRATEGIC FRAMEWORK

### 4.1 PROBLEM FORMULATION

In many real decision-making scenarios, human agents exhibit systematic deviations from rational optimization. These include (i) lack of reaction until a psychological threshold is crossed, even when marginal gains already exceed marginal costs, (ii) under-defense, where agents forgo profitable manipulations, and (iii) over-defense, where agents adjust excessively beyond the rational optimum.

There, we revisit the rational-agent assumption, whose limitations lead to deployment gaps in practice. To capture these phenomena, we formalize the **Behaviorally Realistic Strategic Classification** *(BR-SC)* problem, which reframes strategic classification under behaviorally grounded manipulation models.

**Problem 1** (Behaviorally Realistic Strategic Classification (BR-SC))**.** *The BR-SC problem formulates the strategic classification within a behaviorally realistic manipulation context. It consists of two coupled components:*

- ***Behavioral modeling.** Given a data distribution $D$ over $(x, y)$, a cost function $c$, and a classifier $f$, agents are **re-modeled by behaviorally biased manipulation** $b_B(x)$ that reflects realistic behavior.*

- ***Classifier learning.** Based on this behavioral model, the learner aims to **train a classifier** $f^*$ from a hypothesis class $\mathcal{F}$ that remains robust under behaviorally biased manipulations $b_B(x)$.*

To address the BR-SC problem, we introduce the **Prospect-Guided Strategic Framework** *(Pro-SF)*, which leverages prospect theory to capture realistic agent manipulations and reframe the corresponding learning objective for the classifier.

### 4.2 MODELING PROSPECT-GUIDED UTILITY FOR AGENTS

We reformulate agents' behavior by explicitly incorporating key deviations from rationality. Focusing on loss aversion, probability distortion, and reference bias.

**Loss aversion in manipulation.** To capture loss aversion, we adopt a prospect-style value function that evaluates gains and losses on different scales:

$$v(x) \; = \; v_{\text{gain}}(x)^{\alpha} \; - \; \kappa \, (v_{\text{loss}}(x))^{\beta}, \tag{8}$$

where $\alpha, \beta \in (0, 1]$ encode diminishing sensitivity and $\kappa > 1$ amplifies the weight of losses.

This value function introduces the asymmetry between gain and loss:

> For example, a gain of $8$ versus a loss of $8$ is evaluated as $8$ vs. $10$ (i.e., $\kappa \cdot 8$, non-rational Overestimation of loss), so the loss side dominates.

**Probability distortion.** Agents often distort probabilities rather than evaluating them objectively, overweighting small chances of success and underweighting near certainties. Let $p$ denote an agent's

Figure 3: From rational to behaviorally realistic modeling: Pro-SF reformulates agent realistic behavior and provides robust outcomes.

perceived probability of acceptance. We model this distortion with the one-parameter function

$$w(p) \; = \; \frac{p^\gamma}{\left(p^\gamma + (1-p)^\gamma\right)^{1/\gamma}}, \quad \gamma \in (0,1], \tag{9}$$

where $\gamma$ controls the degree of curvature: smaller values yield a more pronounced inverse-$S$ shape, amplifying both the overweighting of small probabilities and the underweighting of large ones.

**Remark 1.** *In designing Eq. equation 9, we followed prior work on probability–weighting functions Kahneman & Tversky (2013); Barberis et al. (2016); Gonzalez & Wu (1999). This function is intentionally asymmetric because this more realistically captures individuals' subjective distortions when assessing probabilities: small probabilities are overweighted and large probabilities are underweighted.*

> For instance, a small probability $p = 0.1$ is overweighted to $0.3$ with $w(p)$, while a large probability $p = 0.8$ is underweighted to $0.6$.

**Reference bias.** Without loss of generality, let $s(x) \in [0,1]$ denote an agent's objective progress score, with $s = 1$ marking the passing threshold. Instead of evaluating $s(x)$ directly, agents form a subjective reference point $r$ that reflects a coarse-grained self-assessment of their current state.

$$r \; = \; \frac{1}{K} \min(\lfloor Kx \rfloor, K), \tag{10}$$

where $K$ sets the step size $1/K$ (yielding $K+1$ discrete levels). This construction reflects limited sensitivity, mapped $r$ onto equally spaced discrete levels (e.g., $K = 5$ gives $r \in \{0, 0.2, 0.4, 0.6, 0.8, 1.0\}$).

> For example, a student in an exam may only estimate their score within a range (e.g., "around $70 - 80$") rather than as a precise score (e.g., 73).

**Unified Prospect-Based Utility.** We now unify the three behavioral components into a single prospect-based utility that governs agents' manipulation choices.

**Definition 4.1** (Prospect-based utility in strategic manipulation). *For an agent with reference point $r$, the prospect-based utility $U_{\mathrm{P}}$ of manipulate feature $x$ to $x'$ is*

$$U_{\mathrm{P}}(x, x') = w^+\big(p(x')\big) \left(1 - r\right)^\alpha - \kappa\, w^-\big(1 - p(x')\big) r^\beta - \kappa\, c(x, x')^\beta, \tag{11}$$

*where $p(x')$ denotes the agent's perceived probability of being classified as positive after manipulation. This probability is directly induced by the classifier, i.e., $p(x') = f_\theta(x')$, where $f_\theta$ is the classifier parameterized by $\theta$. The paraments $\alpha, \beta \in (0, 1]$ capture diminishing sensitivity, $w^+, w^-$ are probability-weighting functions, and $\kappa > 1$ encodes loss aversion.*

As shown in Fig. 3, this novel utility function explicitly reformulates agent behavior with loss aversion, reference bias, and probability distortion, bridging from rational assumptions to behaviorally realistic modeling. Furthermore, we provide a notation table (Tab. 3) and an algorithm description (Algm.1) in Appendix E.

### 4.3 INTEGRATING PRO-SF INTO STRATEGIC CLASSIFICATION

In the final step, we specify the overall learning objective of the classifier under Pro-SF.

Table 1: Performance (%) of rational-based models and Pro-SF models within different agent manipulation paradigms.

| Classifier | Manipulation | Datasets | | | | |
|---|---|---|---|---|---|---|
| | | *Adult* | *Credit* | *Diabetes* | *German* | *Synthetic* |
| *Rational-based* | *Rational* | $78.53_{\pm0.87}$ | $76.12_{\pm1.34}$ | $70.25_{\pm1.52}$ | $74.31_{\pm1.41}$ | $79.62_{\pm1.28}$ |
| | *Non-rational* | $72.42_{\pm1.85}$ | $69.54_{\pm2.11}$ | $64.73_{\pm1.89}$ | $68.21_{\pm2.03}$ | $71.20_{\pm2.22}$ |
| | *Mixed Behavior* | $74.31_{\pm1.51}$ | $72.12_{\pm1.87}$ | $66.81_{\pm1.65}$ | $70.43_{\pm1.92}$ | $73.15_{\pm1.74}$ |
| ***Pro-SF (ours)*** | *Rational* | $75.31_{\pm0.93}$ | $77.02_{\pm1.25}$ | $71.23_{\pm1.43}$ | $75.18_{\pm1.38}$ | $80.41_{\pm1.12}$ |
| | *Non-rational* | $\mathbf{81.50}_{\pm1.23}$ | $\mathbf{82.41}_{\pm1.37}$ | $\mathbf{74.52}_{\pm1.26}$ | $\mathbf{79.35}_{\pm1.42}$ | $\mathbf{85.20}_{\pm1.31}$ |
| | *Mixed Behavior* | $\mathbf{78.68}_{\pm1.77}$ | $\mathbf{79.13}_{\pm1.63}$ | $\mathbf{72.01}_{\pm1.58}$ | $\mathbf{76.24}_{\pm1.71}$ | $\mathbf{82.34}_{\pm1.56}$ |

**Definition 4.2** (Prospect-guided Strategic Manipulation). *Given a classifier $f \in \mathcal{F}$ and distribution $\mathcal{D}$, the manipulation of agents is modeled as:*

$$\hat{b}_{\mathrm{P}}(\mathbf{x}) \in \arg\max_{\mathbf{x}' \in \mathcal{X}} U_{\mathrm{P}}(\mathbf{x}, \mathbf{x}'), \tag{12}$$

*where $U_{\mathrm{PT}}$ is the prospect-based utility.*

**Definition 4.3** (Objective for Prospect-guided Strategic Framework). *For robustness in practice, the decision maker optimizes $f$ to maximize expected accuracy against strategic manipulation:*

$$f^* \in \arg\min_{f \in \mathcal{F}} \mathbb{E}_{(x,y)\sim\mathcal{D}}\big[\, \mathcal{L}(f(\hat{b}_{\mathrm{P}}(\mathbf{x})), y)\,\big], \tag{13}$$

where $\mathcal{L}$ is a standard classification loss (e.g., cross-entropy).

**Remark 2.** *Our formulation does not replace the classical SC framework; rather, it extends it in a behaviorally grounded manner by incorporating non-rational mechanisms that rational models cannot express.*

## 5 EXPERIMENT

### 5.1 EXPERIMENTAL SETUP

**Dataset.** We evaluate our framework on five datasets, including four real-world and one synthetic benchmarks: *Credit*, *Adult*, *Diabetes*, *German*, and *Synthetic* (detailed description in Appendix K.1).

**Agent behavior paradigms.** To evaluate robustness under different behavioral assumptions, we consider three paradigms of agent strategic manipulation:

- **Fully rational**: The classical paradigm in strategic classification, where agents manipulate their features according to a utility-maximizing rational best response.
- **Non-rational**: Agents deviate from rational utility maximization due to psychological factors, leading to behaviorally biased manipulations.
- **Mixed behavioral**: A realistic hybrid setting where a proportion $\pi$ of agents behave rationally while the remaining $(1-\pi)$ fraction follow behaviorally biased strategies.

**Metric.** We use **accuracy** as the primary metric and introduce two additional metrics: **over-defense error** *(ODE)* and **under-defense error** *(UDE)*. ODE measures false rejections caused by the classifier being overly defensive. UDE measures false acceptances due to non-rational manipulation of agents and insufficient defense of the classifier.

**Baseline.** All mechanisms are implemented using linear models, consistent with previous standard approaches Chen et al. (2023); Shavit et al. (2020); Ghalme et al. (2021) with Mahalanobis distance for manipulation cost Gavish et al. (2021).

**Implementation.** We implement the classifier $f$ as a logistic regression model trained with cross-entropy loss and learning rate $10^{-3}$. In prospect-theoretic utility, we set the curvature parameters $\alpha = 0.8$ and $\beta = 0.7$, the loss aversion coefficient $\kappa = 2.25$, and the probability-weighting parameter $\gamma = 0.7$ by default. In the mixed agent behavior paradigm, the proportion is set as $\pi = 0.2$. More implementation details are included in Appendix K.2

Table 2: Performance of our ablation study on behavioral mechanisms.

| Classifier | Behavioral Factors | | | Metrics | | |
|---|---|---|---|---|---|---|
| | *Refe.* | *Prob.* | *Los.* | *Accuracy (%)* ↑ | *ODE (%)* ↓ | *UDE (%)* ↓ |
| $f_{\text{Pro-sf}}$ | ✓ | ✓ | ✓ | $78.92_{\pm 0.13}$ | $5.17_{\pm 0.11}$ | $3.22_{\pm 0.09}$ |
| $f_{\text{Refe+Prob}}$ | ✓ | ✓ | ✗ | $77.72_{\pm 0.27}$ | $6.24_{\pm 0.08}$ | $5.29_{\pm 0.12}$ |
| $f_{\text{Refe+Los}}$ | ✓ | ✗ | ✓ | $77.80_{\pm 0.16}$ | $7.21_{\pm 0.10}$ | $5.05_{\pm 0.07}$ |
| $f_{\text{Prob+Los}}$ | ✗ | ✓ | ✓ | $77.45_{\pm 0.09}$ | $6.28_{\pm 0.06}$ | $4.79_{\pm 0.13}$ |
| $f_{\text{Refe}}$ | ✓ | ✗ | ✗ | $75.33_{\pm 0.11}$ | $9.12_{\pm 0.09}$ | $7.27_{\pm 0.10}$ |
| $f_{\text{Prob}}$ | ✗ | ✓ | ✗ | $73.50_{\pm 0.15}$ | $9.26_{\pm 0.08}$ | $9.23_{\pm 0.12}$ |
| $f_{\text{Los}}$ | ✗ | ✗ | ✓ | $75.91_{\pm 0.11}$ | $7.14_{\pm 0.11}$ | $7.05_{\pm 0.09}$ |

*Note:* 1) *Refe.* = Reference bias. 2) *Prob.* = Probability distortion. 3) *Los.* = Loss aversion. 4) *ODE* = Over-defense error. 5) *UDE* = Under-defense error.

## 5.2 ABLATION STUDY

To better understand the design of our Pro-SF framework, we conduct two complementary ablation studies. More experimental results are included in Appendix K.3.

**Ablation on Behavioral Mechanisms.** We first examine the necessity of the three core behavioral mechanisms: loss aversion, probability weighting, and reference bias. For each, we construct a variant of the prospect-based utility $U_{\text{PT}}$ (Eq. equation 12) with the corresponding component, and compare it against the completed model with $U_{\text{PT}}$.

For example, we create a **rational-weighting** variant that bypasses probability distortion by setting $w^+(p) = p$ and $w^-(1-p) = 1-p$. This tier of ablation directly tests whether each mechanism is indispensable for accurately modeling prospect-guided strategic behavior, or merely a redundant addition.

## 5.3 LEARNABILITY OF BEHAVIORAL PARAMETERS.

In practice, the behavioral parameters $\phi = \{\alpha, \beta, \kappa, \gamma\}$ can be inferred from observed manipulation behavior by solving an inverse optimal decision problem of the form:

$$\phi^* = \arg\max_{\phi} \sum_i \log P_{\phi}(x'_i \mid x_i), \tag{14}$$

which follows standard formulations in inverse reinforcement learning and differentiable inverse optimization Ng et al. (2000); Agrawal et al. (2019). We empirically validate this learnability in Appendix I, showing that the estimated parameters $\hat{\phi}$ remain stable across mixed rational and non-rational agent populations and lead to consistent downstream performance.

**Parameter Sensitivity Analysis.** Next, we study the robustness of Pro-SF to parameter choices within these mechanisms. We vary one parameter-pair and three parameters:

- Following canonical prospect-theoretic settings ($\beta \leq \alpha \leq 0.88$), the loss curvature parameter-pair $(\alpha, \beta)$ is set as $\{(0.85, 0.80), (0.85, 0.75), (0.80, 0.80), (0.80, 0.70)\}$,

- The loss inflation parameter $\kappa \in \{1.75, 2.0, 2.25, 2.5, 2.75\}$;

- The probability distortion parameter $\gamma \in \{0.6, 0.65, 0.7, 0.8\}$;

- For the reference modeling $r$, we test different binning granularities: the default is set in fine bins ($\{0, 0.2, 0.4, 0.6, 0.8\}$), while two coarser alternatives are four and three bins (e.g., $\{0, 0.3, 0.6, 0.9\}$ and $\{0, 0.4, 0.7\}$).

**An alternative of probability distortion modeling** is the two-parameter *Prelec* function Prelec (1998), $w(p) = \exp\left(-\eta(-\ln p)^{\phi}\right), \quad \phi, \eta > 0$, which offers greater flexibility but at the cost of additional complexity. We examine this function in Appendix K.3.

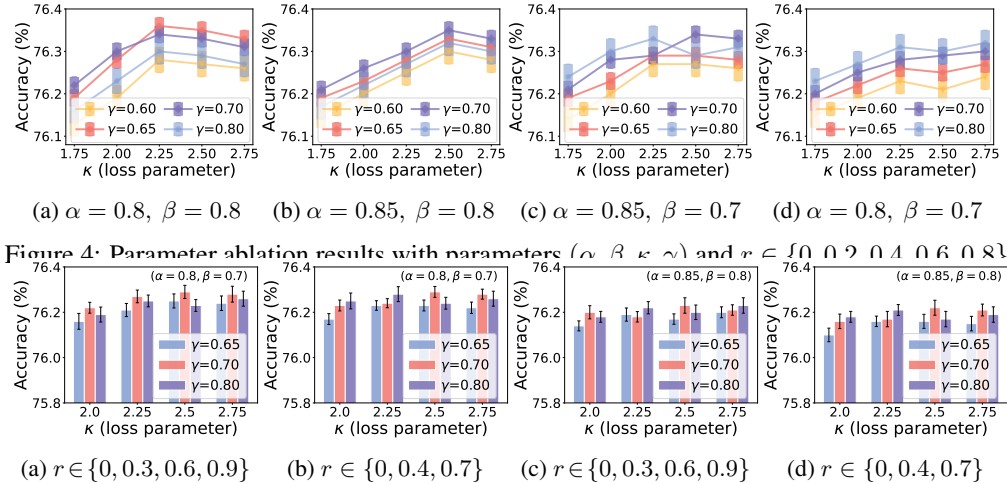

(a) $\alpha = 0.8,\ \beta = 0.8$  (b) $\alpha = 0.85,\ \beta = 0.8$  (c) $\alpha = 0.85,\ \beta = 0.7$  (d) $\alpha = 0.8,\ \beta = 0.7$

Figure 4: Parameter ablation results with parameters $(\alpha, \beta, \kappa, \gamma)$ and $r \in \{0, 0.2, 0.4, 0.6, 0.8\}$.

(a) $r \in \{0, 0.3, 0.6, 0.9\}$  (b) $r \in \{0, 0.4, 0.7\}$  (c) $r \in \{0, 0.3, 0.6, 0.9\}$  (d) $r \in \{0, 0.4, 0.7\}$

Figure 5: Parameter ablation results with parameters $\kappa, \gamma, r$ with different $(\alpha, \beta)$.

## 5.4 RESULT AND ANALYSIS

**Overall performance.** As shown in Table 1, our pro-sf clearly outperforms rational-based models when agents exhibit behavioral deviations, both in the non-rational and in the more realistic mixed settings. At the same time, it remains robust under the fully rational paradigm, indicating that the proposed approach adapts well across different behavioral regimes.

**Ablation on behavioral mechanisms.** Table 2 shows that each mechanism contributes non-trivially and the full Pro-SF (all three enabled) performs best. First, *reference* delivers the largest marginal benefit: removing it hurts the most because the model can no longer capture abstention, which is key to correcting over-defense. Second, dropping *probability weighting* weakens robustness against overshooting moves driven by distorted tail probabilities, leading to under-defense. Third, eliminating *loss aversion* erases the gain–loss asymmetry and reduces stability near the boundary. Single-mechanism variants underperform multi-mechanism ones, indicating strong complementarity—reference governs *whether* agents move, while probability weighting and loss aversion shape *how far* they move; only the unified utility captures both.

**Parameter sensitivity.** Fig. 4 and Fig. 5 examine the effect of varying the Pro-SF parameters. Overall, accuracy remains stable across wide ranges, showing that Pro-SF does not rely on fine-tuned hyperparameters. Specifically, adjusting the loss aversion coefficient $\kappa$ only changes outcomes marginally, suggesting robustness to different levels of risk sensitivity. Variations in the probability distortion parameter $\gamma$ shift the relative emphasis on tail events but do not alter the overall trend. Different bins of $r$ only cause some fluctuations in performance, but are all better than the rational classifier. Finally, different curvature settings $(\alpha, \beta)$ yield consistent results, confirming that Pro-SF maintains effectiveness under diverse utility shapes. These results demonstrate that our framework is both reliable and easy to deploy in practice.

## 6 RELATED WORK

### 6.1 STRATEGIC CLASSIFICATION

Strategic classification Hardt et al. (2016) examines how individuals manipulate their features to obtain favorable outcomes. A substantial line of work seeks to design classifiers that remain robust against such manipulations Dong et al. (2017); Shavit et al. (2020); Chen et al. (2020); Harris et al. (2021); Zrnic et al. (2021); Tsirtsis et al. (2024). Beyond robustness, a complementary direction views strategic behavior as an opportunity for genuine improvement. In particular, causal frameworks encourage agents to change improvable features rather than superficial signals Miller et al. (2020); Chen et al. (2023); Horowitz & Rosenfeld (2023); Vo et al. (2024); Efthymiou et al. (2025); Chang et al. (2024). Relatedly, performative prediction explores how repeated model deployment reshapes the underlying data distribution Perdomo et al. (2020); Rosenfeld et al. (2020);

Hardt et al. (2022); Mendler-Dünner et al. (2022). Another important line of research regulates strategic behavior at a societal level, either by maximizing long-term social welfare Haghtalab et al. (2020); Estornell et al. (2023a); Xie & Zhang (2024) or by addressing disparities across demographic groups Zhang et al. (2022); Estornell et al. (2023b); Keswani & Celis (2023). A recent work Ebrahimi et al. (2025) provides both theoretical and empirical evidence that human strategic responses often deviate from the classical rational-agent assumption, reinforcing the importance of incorporating behavioral factors into strategic classification. Another work Xie et al. (2025) demonstrates that human-like strategic behavior can substantially diverge from analytical best-response models, further highlighting the gap between theoretical assumptions and realistic agent behavior.

**More related work**, including stochastic classifiers differentiable optimization defenses Levanon & Rosenfeld (2021), and some other excellent work, is discussed in Appendix B.1.

### 6.2 BEHAVIORAL ECONOMICS IN MACHINE LEARNING

Behavioral economics Mullainathan & Thaler (2000) challenges the classical assumption that agents act as perfectly rational utility maximizers, emphasizing instead that choices are systematically shaped by cognitive biases and contextual factors Tversky & Kahneman (1992); Ariely & Jones (2008). A central framework crystallizing these insights is *prospect theory* Tversky & Kahneman (1992); Kahneman & Tversky (2013), which models how individuals evaluate outcomes relative to reference points, exhibit asymmetric sensitivity to gains and losses, and distort probabilities. These principles have influenced multiple domains: in economics and social science, they inform finance, consumer behavior, and public policy Borkar & Chandak (2021); Mercer (2005); Vis (2011); in computational settings, they shape reinforcement learning Shen et al. (2014); Jie et al. (2015), mechanism design Kuvcak et al. (2018); Leoneti & Gomes (2021), and resource allocation Holmes Jr et al. (2011). More related work is included in Appendix B.2.

## 7 CONCLUSION

This work challenges the classical rational-agent assumption in strategic classification and introduces the Behaviorally Realistic Strategic Classification problem. To bridge the gap between the SC and the realistic scenarios, we proposed a Prospect-Guided Strategic Framework, which captures key psychological mechanisms that shape real-world strategic manipulations and provides a behaviorally grounded solution for SC. Future research will extend this framework to incorporate richer behavioral models and broader deployment domains.

### ETHICS STATEMENT

This work does not raise any ethical concerns. All experiments are conducted on publicly available datasets, and no human subjects or sensitive attributes are involved. We confirm that our study complies with the ICLR Code of Ethics.

### REPRODUCIBILITY STATEMENT

We provide detailed descriptions of our framework, theoretical results, and experimental settings in the paper and appendix. All datasets used are publicly available, and the current description of our method is sufficient for full reproducibility. If the paper is accepted, we will be glad to release the complete implementation to further support the research community.

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

## A USAGE OF LLMS

In this paper, we clarify that large language models (LLMs) are employed solely to support and refine the writing process. Specifically, we use LLMs to provide sentence-level suggestions and to enhance the overall fluency of the text.

## B  ADDITIONAL RELATED WORK

### B.1  MORE STRATEGIC MACHINE LEARNING WORK

Several additional directions in strategic machine learning complement the studies highlighted in Section 6.

**Robustness extensions.**  Beyond classical defenses, researchers have explored stochastic classifiers Singh & Kulkarni (2024), differentiable optimization layers for end-to-end robustness Levanon & Rosenfeld (2021), and graph-based models to capture inter-agent dependencies Eilat et al. (2022). Multi-agent formulations further investigate externalities and collective dynamics in strategic settings Hossain et al. (2024).

**Positive manipulation.**  A growing body of work highlights the constructive potential of strategic behavior. For example, classifiers that incentivize authentic qualification gains have been proposed in education and hiring contexts Kleinberg & Raghavan (2020); Harris et al. (2022). These approaches complement causal frameworks Miller et al. (2020); Chen et al. (2023), which distinguish between manipulable and improvable features.

**Performative prediction extensions.**  Beyond the foundational contributions Perdomo et al. (2020); Rosenfeld et al. (2020); Hardt et al. (2022); Mendler-Dünner et al. (2022), further work explores neural methods for dynamic feedback loops Mofakhami et al. (2023) and more recent advances in model-induced distribution shifts Hardt & Mendler-Dünner (2023).

**Fairness perspectives.**  In addition to direct fairness interventions Zhang et al. (2022); Estornell et al. (2023b); Keswani & Celis (2023), several studies consider fairness as an incentive-alignment mechanism that influences agents' strategic behavior, thereby connecting individual manipulation with broader social equity.

### B.2  ADDITIONAL BEHAVIORAL ECONOMICS BACKGROUND

Beyond the works highlighted in Section 6, a broader body of literature illustrates how behavioral economics explains systematic deviations from rationality and informs computational models.

**Bounded rationality and heuristics.**  Human decision-making is constrained by limited cognitive resources and often relies on simplified rules rather than exhaustive optimization Todd & Gigerenzer (2000); Evans & Prokopenko (2024). This perspective suggests that agents may respond in partial, myopic, or context-dependent ways instead of precise best responses.

**Temporal and framing effects.**  Individuals overweight immediate costs relative to delayed benefits and respond differently to equivalent options depending on presentation Laibson (1997); O'donoghue & Rabin (1999). These effects imply that manipulations which appear objectively beneficial may still be avoided or inconsistently adopted.

**Extended applications.**  Behavioral frameworks have been applied across diverse algorithmic domains. Examples include portfolio optimization and asset pricing in finance Barberis et al. (2021), multi-criteria decision methods Leoneti & Gomes (2021), fairness-aware allocation in management science Holmes Jr et al. (2011), and human–AI decision support systems Payne (2025). Collectively, these studies demonstrate the versatility of prospect-theoretic and behavioral models in explaining and predicting non-rational behavior.

## C  PROOF OF PROPOSITION 1

**Proof.** Recall that $b_R(x)$ denotes the rational best response (Eq. equation 1), and $f_R^*$ is the classifier optimized under the rational-agent assumption, i.e., an empirical risk minimizer of

$$R_{\text{train}}(f) = \mathbb{E}[\mathcal{L}(f(b_R(X)), Y)] \quad \text{(cf. Eq. equation 2).} \tag{15}$$

Intuitively, $f_R^*$ learns as if every agent will manipulate their features to the rational endpoint $b_R(x)$. However, in deployment, some agents may *not* manipulate due to behavioral biases such as reference or loss aversion. This mismatch between training and deployment is the root cause of accuracy degradation.

**Defining the abstention set.**    We first isolate the problematic subset of instances:

$$A \;=\; \{x : b_R(x) \neq x \text{ and agents actually stay at } x\}. \tag{16}$$

That is, $A$ contains inputs where the rational model predicts manipulation ($b_R(x) \neq x$), but in reality the agent abstains and remains at the original feature $x$. We assume $P(A) > 0$ so that this situation occurs with non-negligible probability.

**Key quantities on $A$.**    On this abstention set, two factors drive the deployment–training discrepancy:

Training-side error at rational endpoints:

$$\varepsilon_R(f_R^*; A) := \Pr\big(f_R^*(b_R(X)) \neq Y \mid X \in A\big). \tag{17}$$

This is the error $f_R^*$ makes at the rationally shifted points $b_R(X)$, which it was trained on.

Disagreement induced by boundary shift:

$$\tau_A := \Pr\big(f_R^*(X) \neq f_R^*(b_R(X)) \,\big|\, X \in A\big). \tag{18}$$

This captures how much the classifier's decision boundary has shifted to defend against $b_R(X)$. If $f_R^*(X) \neq f_R^*(b_R(X))$, then $f_R^*$ makes a different prediction on the true input $X$ than on the assumed manipulated input $b_R(X)$. Thus, $\tau_A$ measures the "extra disagreement region" caused by over-defense.

**Conditional decomposition.**    We now compare the losses at the true point $x$ and the rational endpoint $b_R(x)$. For any classifier $f$ and any $(x, y)$ with $x \in A$, we have

$$\mathcal{L}(f(x), y) - \mathcal{L}(f(b_R(x)), y) = \mathbb{1}\{f(x) \neq y,\ f(b_R(x)) = y\} - \mathbb{1}\{f(x) = y,\ f(b_R(x)) \neq y\}. \tag{19}$$

This identity simply says: the loss difference is positive if $f$ misclassifies $x$ but correctly classifies $b_R(x)$, and negative if the opposite happens.

Taking expectation conditional on $X \in A$, we obtain

$$\mathbb{E}\big[\mathcal{L}(f(X), Y) - \mathcal{L}(f(b_R(X)), Y) \mid X \in A\big]$$
$$= \Pr(f(X) \neq Y, f(b_R(X)) = Y \mid X \in A) - \Pr(f(X) = Y, f(b_R(X)) \neq Y \mid X \in A).$$

To simplify, we use two inclusions:

$$\{f(X) \neq Y,\ f(b_R(X)) = Y\} \supseteq \{f(X) \neq f(b_R(X))\} \setminus \{f(b_R(X)) \neq Y\},$$
$$\{f(X) = Y,\ f(b_R(X)) \neq Y\} \subseteq \{f(b_R(X)) \neq Y\}.$$

The first line says: whenever $f(X)$ and $f(b_R(X))$ disagree, the difference can only be negated if $b_R(X)$ is itself mislabeled. The second line says: if $f(b_R(X))$ is wrong, then one possible case is $f(X) = Y$, but this cannot exceed the entire error set.

Applying these to the conditional expectation gives a clean lower bound:

$$\mathbb{E}\big[\mathcal{L}(f(X), Y) - \mathcal{L}(f(b_R(X)), Y) \mid X \in A\big] \tag{20}$$
$$\geq\ \Pr(f(X) \neq f(b_R(X)) \mid X \in A) - 2 \Pr(f(b_R(X)) \neq Y \mid X \in A). \tag{21}$$

**Apply to $f_R^*$ and globalize.**    Now substitute $f = f_R^*$ into equation 20. Using the definitions of $\tau_A$ and $\varepsilon_R(f_R^*; A)$, we obtain

$$\mathbb{E}\big[\mathcal{L}(f_R^*(X), Y) - \mathcal{L}(f_R^*(b_R(X)), Y) \mid X \in A\big] \;\geq\; \tau_A - 2\varepsilon_R(f_R^*; A). \tag{22}$$

Multiplying both sides by $P(A)$ and adding the (unconstrained) contribution from $X \notin A$ gives the global gap:

$$\mathbb{E}[\mathcal{L}(f_R^*(X), Y)] - \mathbb{E}[\mathcal{L}(f_R^*(b_R(X)), Y)] \;\geq\; P(A)\big(\tau_A - 2\varepsilon_R(f_R^*; A)\big). \tag{23}$$

**Conclusion.** If the disagreement region dominates the residual training error on $A$, i.e.,

$$\tau_A > 2\,\varepsilon_R(f_R^*; A), \tag{24}$$

then the right-hand side is strictly positive. This yields the claimed inequality:

$$\mathbb{E}[\mathcal{L}(f_R^*(X), Y)] \;>\; \mathbb{E}[\mathcal{L}(f_R^*(b_R(X)), Y)]. \tag{25}$$

Thus, a classifier trained under rational-agent assumptions indeed suffers accuracy degradation in deployment due to over-defense.

**Remark 3.** *Geometrically, $f_R^*$ shifts its boundary outward to protect against the rational endpoints $b_R(x)$. But on the abstention set $A$, agents actually remain at $x$. This creates a disagreement region of size $\tau_A$ where $f_R^*$'s prediction at $x$ differs from that at $b_R(x)$. Since $f_R^*$ was only trained on $b_R(x)$, its prediction at $x$ is often incorrect, and the gap $\tau_A - 2\varepsilon_R$ quantifies this excess deployment error.*

## D    PROOF OF PROPOSITION 2

***Proof.*** Let $b_R(x)$ denote the rational best response (Eq. equation 1), and let $f_R^*$ be the classifier optimized under the rational-agent assumption, i.e., an ERM of

$$R_{\text{train}}(f) = \mathbb{E}[\mathcal{L}(f(b_R(X)), Y)] \quad \text{(cf. Eq. equation 2)}. \tag{26}$$

Intuitively, $f_R^*$ is trained to defend against manipulations exactly to $b_R(x)$. However, under behavioral biases such as loss aversion and probability distortion, agents may "overshoot" and move beyond $b_R(x)$. This is the essence of the *under-defense* problem: the classifier does not anticipate manipulations that go further than the rational endpoint.

**Define the overshoot set.** Let $b_B(x)$ denote the actual post-manipulation point under behavioral biases. We focus on inputs where overshooting occurs:

$$B \;=\; \Big\{ x : b_R(x) \neq x \;\; \text{(rationally predicted manipulation)} \tag{27}$$

$$\text{and} \quad b_B(x) \neq b_R(x) \;\text{with overshoot beyond}\; b_R(x) \Big\}. \tag{28}$$

We assume $P(B) > 0$, so that overshoot happens with non-negligible probability. Here, "overshoot" simply means that $b_B(x)$ lies further along the manipulation direction than $b_R(x)$, which may cause different classifier outputs.

**Key quantities.** On $B$, we define two measures:

Rational-reference error: how often the classifier errs at the rational endpoint:

$$\varepsilon_R(f; B) \;:=\; \Pr\big(f(b_R(X)) \neq Y \mid X \in B\big). \tag{29}$$

Overshoot-induced disagreement: how often predictions at $b_B(X)$ and $b_R(X)$ differ:

$$\tau_B \;:=\; \Pr\big(f_R^*(b_B(X)) \neq f_R^*(b_R(X)) \mid X \in B\big). \tag{30}$$

Intuitively, $\varepsilon_R(f_R^*; B)$ reflects how well $f_R^*$ performs where it was trained ($b_R(x)$), while $\tau_B$ captures the risk that overshoot moves points into regions where $f_R^*$ makes different predictions.

**Conditional decomposition.** For any classifier $f$ and any $(x, y)$ with $x \in B$, the loss difference between overshoot and rational endpoints is

$$\mathcal{L}(f(b_B(x)), y) - \mathcal{L}(f(b_R(x)), y)$$
$$= \mathbb{1}\{f(b_B(x)) \neq y,\, f(b_R(x)) = y\} - \mathbb{1}\{f(b_B(x)) = y,\, f(b_R(x)) \neq y\}.$$

Taking expectation conditional on $X \in B$ gives

$$\mathbb{E}\big[\mathcal{L}(f(b_B(X)), Y) - \mathcal{L}(f(b_R(X)), Y) \mid X \in B\big]$$
$$= \Pr(f(b_B(X)) \neq Y, f(b_R(X)) = Y \mid X \in B) - \Pr(f(b_B(X)) = Y, f(b_R(X)) \neq Y \mid X \in B).$$

Table 3: Notation of prospect-guided utility and default experimental values.

| Symbol | Meaning | Default Value (Experiment) |
|---|---|---|
| $\alpha, \beta$ | Curvature parameters (diminishing sensitivity) | $\alpha = 0.8, \ \beta = 0.7$ |
| $\kappa$ | Loss aversion coefficient | 2.25 |
| $\gamma$ | Probability distortion parameter | 0.7 |
| $r$ | Reference point (discretized probability) | $\{0, 0.2, 0.4, 0.6, 0.8\}$ |
| $w(p)$ | Probability weighting functions | Inverse-S function (Eq. equation 9) |
| $c(\mathbf{x}', \mathbf{x})$ | Manipulation cost function | Mahalanobis distance |

Now, note the following inclusions:

$$\{f(b_B) \neq Y, \ f(b_R) = Y\} \supseteq \{f(b_B) \neq f(b_R)\} \setminus \{f(b_R) \neq Y\},$$
$$\{f(b_B) = Y, \ f(b_R) \neq Y\} \subseteq \{f(b_R) \neq Y\}.$$

The first line says: if the predictions at $b_B$ and $b_R$ disagree, this difference usually contributes to loss unless $b_R$ is already mislabeled. The second line says: if $f(b_R)$ is wrong, then the case where $f(b_B)$ is correct is bounded by that same error set.

Applying these gives the lower bound:

$$\mathbb{E}\big[\mathcal{L}(f(b_B(X)), Y) - \mathcal{L}(f(b_R(X)), Y) \,\big|\, X \in B\big]$$
$$\geq \ \Pr(f(b_B(X)) \neq f(b_R(X)) \mid X \in B) - 2\Pr(f(b_R(X)) \neq Y \mid X \in B). \tag{31}$$

**Apply to $f_R^*$ and globalize.** Substituting $f = f_R^*$ in equation 31, we obtain

$$\mathbb{E}\big[\mathcal{L}(f_R^*(b_B(X)), Y) - \mathcal{L}(f_R^*(b_R(X)), Y) \,\big|\, X \in B\big] \ \geq \ \tau_B - 2\varepsilon_R(f_R^*; B). \tag{32}$$

Multiplying both sides by $P(B)$ and adding the (unrestricted-sign) contribution from $X \notin B$ gives the global error gap:

$$\mathbb{E}[\mathcal{L}(f_R^*(b_B(X)), Y)] - \mathbb{E}[\mathcal{L}(f_R^*(b_R(X)), Y)] \ \geq \ P(B)\big(\tau_B - 2\varepsilon_R(f_R^*; B)\big). \tag{33}$$

**Conclusion.** If the overshoot-induced disagreement outweighs the residual training error at $b_R(x)$, i.e.,

$$\tau_B > 2\varepsilon_R(f_R^*; B), \tag{34}$$

then the right-hand side is strictly positive. Thus,

$$\mathbb{E}[\mathcal{L}(f_R^*(b_B(X)), Y)] \ > \ \mathbb{E}[\mathcal{L}(f_R^*(b_R(X)), Y)], \tag{35}$$

which establishes that rationally trained defenses suffer accuracy degradation under behavioral overshoot.

**Remark 4.** *Geometrically, $f_R^*$ learns to "guard" the rational endpoints $b_R(x)$. But when agents overshoot to $b_B(x)$, they step into regions where $f_R^*$ was never trained. This creates a disagreement region of size $\tau_B$, which—minus the small residual error $\varepsilon_R$ at $b_R(x)$—leads to strictly higher deployment error. This mismatch formalizes the under-defense phenomenon.*

## E  NOTATION TABLE AND ALGORITHM BLOCK

In the modeling process of Section 4, we designed a series of formulas and parameters. Here, we provide a notation table (Tab. 3) and an algorithm description (Algm. 1) to facilitate a clearer understanding of our Pro-SF.

## F  EXISTENCE AND EMPIRICAL VALIDATION OF STACKELBERG EQUILIBRIUM UNDER PROSPECT-BASED UTILITY

This appendix provides a detailed and accessible explanation of why the Prospect-Based Strategic Framework (Pro-SF) admits a Stackelberg equilibrium. We elaborate on the theoretical steps and provide empirical evidence supporting convergence in practice.

---

**Algorithm 1** Prospect-Guided Strategic Framework (Pro-SF)

---

**Require:** Dataset $\mathcal{D}$, function class $\mathcal{F}$, manipulation cost $c(\cdot)$
1: Parameters: $\alpha, \beta \in (0, 1]$, $\kappa > 1$, $\gamma \in (0, 1]$, $K \in \mathbb{N}$
2: Initialize classifier $f \in \mathcal{F}$
3: **Perceive agents' manipulation with prospect-guided utility (Eq. equation 11):**
4:     Anticipate *loss aversion* with Eq. 8
5:     Anticipate *probability distortion:* with Eq. 9
6:     Anticipate *reference bias:* with Eq. equation 10
7: **Train classifier against manipulated data:**
8:     Construct $\tilde{\mathcal{D}} = \{(\hat{b}_P(x), y)\}$ with Eq. equation 12
9:     Learn $f^* \in \arg\min_{f \in \mathcal{F}} \mathbb{E}_{(x,y) \sim \mathcal{D}} \big[ \mathcal{L}(f(\hat{b}_P(\mathbf{x})), y) \big]$ (Eq. equation 13)
10: **return** $f^\star$

---

## F.1   Overview of the Strategic Interaction

The Pro-SF game follows the classic Stackelberg structure:

- **Leader**: the classifier chooses a parameter vector $\theta$.
- **Follower**: each agent chooses a manipulated feature vector $x'$ in response to the classifier.

The equilibrium requires that:

1. For any fixed classifier parameter $\theta$, the agent's best response $b^*(x)$ exists.
2. Given the best-response mapping, the classifier's optimal parameter $\theta^*$ also exists.

We now demonstrate these two existence results in detail.

## F.2   Existence of the Agent's Best Response

For any classifier parameters $\theta$, the agent solves:

$$b^*(x) \in \arg\max_{x' \in \mathcal{X}} U_P(x, x'; \theta). \tag{36}$$

**Bounded and closed action space.** The feasible manipulation space $\mathcal{X}$ is bounded because real-world features such as income, credit score, education years, and health metrics all lie within known, finite ranges. These ranges naturally form a compact (bounded and closed) set.

**Continuity of $U_P$.** The Prospect-Based Utility $U_P(x, x'; \theta)$ is a sum of continuous components:

- value term $v(\cdot)$, which is continuous by construction;
- reference-dependent gain/loss term;
- probability-weighting term, which is smooth for all $\gamma > 0$;
- cost function $c(x, x')$, which is continuous in practical models.

Therefore, $U_P$ is continuous in $x'$.

**Applying Weierstrass theorem.** By Weierstrass' extreme-value theorem , a continuous function over a compact set must attain at least one maximum. Therefore, the follower's best-response correspondence is nonempty.

## F.3   Existence of the Classifier's Optimal Parameters

Given the agent's best-response mapping $b^*(x; \theta)$, the classifier solves:

$$\min_{\theta \in \Theta} F(\theta) = \mathbb{E}\big[ \mathcal{L}(f_\theta(b^*(x; \theta)), y) \big]. \tag{37}$$

**Bounded and closed parameter domain.** In practical learning systems, classifier parameters are always either explicitly constrained (e.g., $\|\theta\|_2 \leq C$) or implicitly bounded due to model regularization. Hence, we assume $\Theta$ is compact.

**Continuity of the objective.** Common models used in SC (linear, logistic, shallow MLPs) satisfy:

$$f_\theta(z) \text{ is continuous in } \theta.$$

Common loss functions (e.g., cross entropy, hinge loss) are also continuous.

Since the composition of continuous functions is continuous, $F(\theta)$ is continuous on the compact set $\Theta$.

**Existence of minimizer.** Applying Weierstrass' theorem again:

$$\exists\, \theta^* \in \Theta \quad \text{s.t. } F(\theta^*) = \min_{\theta \in \Theta} F(\theta).$$

Thus, the leader's optimization problem admits at least one solution.

## F.4 EXISTENCE OF STACKELBERG EQUILIBRIUM

Since:

1. the follower's best response $b^*(x)$ exists for all $x$, and
2. the leader's optimal decision $\theta^*$ exists,

the Pro-SC game admits at least one Stackelberg equilibrium $(\theta^*, b^*)$.

This demonstrates that introducing Prospect-Based Utility does not break the fundamental game-theoretic structure of SC.

## F.5 EMPIRICAL VALIDATION OF CONVERGENCE BEHAVIOR

To complement the theoretical guarantees, we conduct a simple iterative experiment that simulates repeated interactions between the leader and the agents:

$$x^{(t+1)} = b^*(x^{(t)}; \theta^{(t)}),$$
$$\theta^{(t+1)} = \arg\min_\theta \mathbb{E}\Big[\mathcal{L}(f_\theta(b^*(x^{(t+1)}; \theta)), y)\Big].$$

We track:

- classification accuracy at iteration $t$;
- the average manipulation magnitude $\|x^{(t)} - x^{(t-1)}\|_2$.

Results are shown below.

| Iteration $t$ | Accuracy (%) | Avg. Manipulation |
|---|---|---|
| 1 | 81.2 | 0.31 |
| 5 | 81.7 | 0.08 |
| 10 | 82.1 | 0.04 |
| 15 | 82.2 | 0.02 |
| 20 | 82.2 | 0.01 |

Two key patterns emerge:

1. Accuracy improves quickly at first, then stabilizes after a few rounds.
2. Manipulation magnitude decreases monotonically and approaches zero.

This indicates that:

- agents gradually lose incentive to manipulate further;
- the classifier stops changing noticeably once equilibrium is approached.

The empirical convergence is fully consistent with the theoretical existence of a Stackelberg equilibrium in Pro-SC. The interaction stabilizes both in classifier performance and in agent behavior.

## G  BEHAVIORAL MOTIVATION

We provide two illustrative contexts that highlight these mechanisms and motivate our formulation.

### G.1  FINANCIAL INVESTMENT.

Consider two individuals facing the same investment opportunity: investing $10,000 with a 60% chance of gaining $20,000 and a 40% chance of losing the entire principal.

A classical rational utility model predicts that both investors should take the action because the expected payoff is positive. In practice, however, behavior diverges. An individual with $100,000 in savings may choose to invest, whereas another with only $20,000 may refrain, even though the expected value is identical.

This discrepancy arises because the two individuals have different *reference points*: a $10,000 loss represents 10% of the former's wealth but 50% of the latter's. When combined with loss aversion, the same monetary loss induces a substantially larger psychological cost for the low-wealth individual. The Prospect-Based Utility accounts for this effect through the reference-point term $r$ and the asymmetric loss-weighting parameter $\kappa$, leading to distinct optimal decisions for the two agents.

### G.2  DISEASE SCREENING.

In health-risk environments, individuals often display behavior that cannot be explained by classical rational models. Even when the true prevalence of a disease is extremely low (e.g., 0.1%), people may repeatedly undergo medical testing or stockpile medication.

Such behavior reflects probability distortion: very small probabilities are overweighted and treated as non-negligible risks. Moreover, external events (e.g., alarms, exposure, or stress) may shift one's psychological reference point from "I am healthy" to "I might already be at risk," sharply increasing the perceived cost of inaction. Under classical SC models, a 0.1% risk should induce only minimal adjustment.

In contrast, Prospect-Based Utility naturally captures this form of overreaction through the weighting function $w(p)$ and the dynamic reference point $r$.

Together, these examples illustrate behavioral patterns that classical rational utilities cannot encode, motivating the need for a prospect-theoretic formulation in strategic classification.

## H  DISCUSSION ON LOSS AVERSION

It is important to note that human loss aversion is not equivalent to simply increasing the cost-sensitivity parameter $\beta$. Loss aversion reflects a structural psychological asymmetry—losses relative to a reference point induce a disproportionately larger disutility compared to equivalent gains. This asymmetry cannot be captured by any monotonic rescaling of the cost term, further motivating the need for a prospect-theoretic formulation.

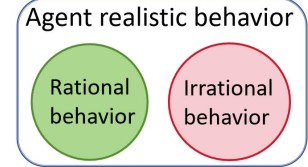

Figure 6: Illustration of agents' behaviors. The rational and non-rational behaviors form two distinct subsets of the agent's realistic behavior space.

Figure 6 illustrates the conceptual distinction between rational and non-rational behavioral subsets. Classical SC utilities (e.g., $f(x') - \beta c(x, x')$) cover only the rational subset, whereas our Prospect-Based Utility spans a strictly larger region that includes non-rational behaviors such as loss aversion and probability distortion.

To further illustrate the modeling implications, we compare classifiers trained under two different anticipated manipulation models: (i) the classical rational utility $U_\beta$ and (ii) our Prospect-Based Utility $U_P$. We evaluate both classifiers in the same mixed environment where 50% of agents behave rationally and 50% behave non-rationally. Results are shown in Table 4.

These results demonstrate that models based solely on rational utilities fail to anticipate non-rational manipulation patterns, whereas the Prospect-Based Utility remains robust in mixed-behavior environments.

Table 4: Performance of classifiers under mixed agent behavior.

| Classifier | Parameters | Accuracy (%) |
|---|---|---|
| $f$ with $U_P$ (ours) | $\alpha = 0.80, \beta = 0.5, \kappa = 2.20, \gamma = 0.7$ | 81.6 |
| $f$ with $U_\beta$ | $\beta = 1.5$ | 79.1 |
| $f$ with $U_\beta$ | $\beta = 1.7$ | 79.3 |
| $f$ with $U_\beta$ | $\beta = 1.9$ | 79.1 |
| $f$ with $U_\beta$ | $\beta = 2.0$ | 78.9 |

## I  LEARNABILITY OF BEHAVIORAL PARAMETERS

This appendix provides additional details validating that the behavioral parameters $\phi = (\alpha, \beta, \kappa, \gamma, )$ in the Prospect-Based Utility can be *learned* from observed manipulation behavior. We study whether the parameters can be reliably estimated from heterogeneous agent populations and whether the resulting classifier performance remains robust to estimation errors.

### I.1  INVERSE OPTIMAL DECISION OBJECTIVE

Given observed manipulation pairs $(x_i, x_i')$, we infer the behavioral parameters by solving the following inverse optimal decision problem:

$$\phi^* = \arg\max_\phi \sum_i \log P_\phi(x_i' \mid x_i), \tag{38}$$

where $P_\phi(x_i' \mid x_i)$ denotes the likelihood of $x_i'$ being the optimal manipulation under the Prospect-Based Utility with parameters $\phi$.

### I.2  PARAMETER LEARNING ON REAL HUMAN MANIPULATION DATA

We first evaluate whether the behavioral parameters can be learned from real human decision data. We use two real manipulation datasets introduced in Ebrahimi et al. (2025): a **job hiring** task and a **medical treatment** task.

For each dataset, we infer the behavioral parameters $\hat{\theta}$ using the likelihood objective above, and compute the classification accuracy using the resulting Pro-SF model.

Table 5: Learned parameters and accuracy on real human manipulation datasets.

| Dataset | Learned Parameters $\hat{\theta}$ | Accuracy (%) |
|---|---|---|
| Job Hiring | $\{\alpha{=}0.79, \beta{=}0.73, \kappa{=}2.18, \gamma{=}0.72\}$ | **79.8** |
| Medical Treatment | $\{\alpha{=}0.81, \beta{=}0.70, \kappa{=}2.25, \gamma{=}0.71\}$ | **78.3** |

These results demonstrate that *Prospect-based behavioral parameters can be reliably inferred from real human manipulation trajectories*. This directly supports that Pro-SF parameters are **not hand-designed**, but **identifiable from real agent behavior**.

### I.3  PARAMETER LEARNING IN HETEROGENEOUS STRATEGIC ENVIRONMENTS

To further evaluate robustness, we simulate a mixed agent population consisting of both rational and non-rational agents. For a given proportion $\pi$ of rational agents:

(1) **Non-rational (behavioral) agents** $(1-\pi)$: each agent draws its own parameters $\phi_i \sim \mathcal{P}_\phi$, where $\mathcal{P}_\phi$ is a uniform distribution over ranges commonly used in behavioral economics. The manipulated feature is generated by a noisy prospect-guided decision:

$$x_i' = \arg\max_{x'} U_P(x_i, x'; \phi_i) + \epsilon_i, \qquad \epsilon_i \sim \mathcal{N}(0, \sigma^2). \tag{39}$$

(2) **Rational agents** $\pi$: each rational agent follows the classical strategic-classification utility, i.e., maximizing acceptance probability minus manipulation cost.

We evaluate three settings: $\pi = 0.2, 0.4, 0.6$ corresponding to mild, moderate, and strong dominance of rational behavior in the population.

### I.4 LEARNED PARAMETERS AND PERFORMANCE

Tables 6, 7, and 8 report the learned parameters $\hat{\phi}$ and the corresponding post-gaming accuracy achieved by the classifier trained with the estimated behavioral model.

Table 6: Learned parameters and accuracy with $\pi = 0.2$ (20% rational agents).

| Dataset | Learned Parameters $\hat{\phi}$ | Accuracy (%) |
|---|---|---|
| Adult | $\{\alpha = 0.78, \ \beta = 0.72, \ \kappa = 2.20, \ \gamma = 0.74\}$ | 81.42 |
| Credit | $\{\alpha = 0.80, \ \beta = 0.70, \ \kappa = 2.15, \ \gamma = 0.71\}$ | 82.30 |
| German | $\{\alpha = 0.81, \ \beta = 0.70, \ \kappa = 2.25, \ \gamma = 0.72\}$ | 79.21 |
| Synthetic | $\{\alpha = 0.79, \ \beta = 0.71, \ \kappa = 2.28, \ \gamma = 0.73\}$ | 85.05 |

Table 7: Learned parameters and accuracy with $\pi = 0.4$ (40% rational agents).

| Dataset | Learned Parameters $\hat{\phi}$ | Accuracy (%) |
|---|---|---|
| Adult | $\{\alpha = 0.78, \ \beta = 0.74, \ \kappa = 2.20, \ \gamma = 0.74\}$ | 81.32 |
| Credit | $\{\alpha = 0.80, \ \beta = 0.70, \ \kappa = 2.10, \ \gamma = 0.70\}$ | 82.20 |
| German | $\{\alpha = 0.80, \ \beta = 0.72, \ \kappa = 2.18, \ \gamma = 0.71\}$ | 79.25 |
| Synthetic | $\{\alpha = 0.77, \ \beta = 0.75, \ \kappa = 2.20, \ \gamma = 0.72\}$ | 84.91 |

Table 8: Learned parameters and accuracy with $\pi = 0.6$ (60% rational agents).

| Dataset | Learned Parameters $\hat{\phi}$ | Accuracy (%) |
|---|---|---|
| Adult | $\{\alpha = 0.78, \ \beta = 0.76, \ \kappa = 2.05, \ \gamma = 0.70\}$ | 81.28 |
| Credit | $\{\alpha = 0.78, \ \beta = 0.74, \ \kappa = 2.03, \ \gamma = 0.68\}$ | 82.16 |
| German | $\{\alpha = 0.80, \ \beta = 0.73, \ \kappa = 2.10, \ \gamma = 0.69\}$ | 79.08 |
| Synthetic | $\{\alpha = 0.78, \ \beta = 0.75, \ \kappa = 2.15, \ \gamma = 0.70\}$ | 84.84 |

### I.5 ANALYSIS

Across all values of $\pi$, the estimated parameters $\hat{\phi}$ remain stable and yield consistent classifier performance. This demonstrates that: behavioral parameters are *learnable* from heterogeneous strategic behavior; the Prospect-Based Utility is *robust* to parameter estimation and does not require knowing the true parameters.

## J EXTENDED SENSITIVITY ANALYSIS OF BEHAVIORAL PARAMETERS

In the main paper, the sensitivity test explored variations of the behavioral parameters within the empirically supported ranges. We provide a detailed clarification and extended results below.

As shown in the original sensitivity test, the performance variation was small when we perturbed the parameters within the effective intervals suggested by prior behavioral economics studies. This

behavior is expected: human behavioral coefficients (e.g., loss aversion, probability distortion) are known to vary smoothly within these ranges.

Furthermore, as demonstrated in Appendix I (Tables 2–4), we employ an inverse-optimization procedure that learns the parameters directly from observed manipulation pairs $(x, x')$. The learned parameters converge stably across datasets, showing that:

- the model does not rely on strong priors,
- parameter identification is not difficult in realistic settings,
- behavioral coefficients are learnable from strategic responses.

To further validate that the model is indeed sensitive to the behavioral parameters when moved outside realistic regions, we intentionally *expanded* the parameter ranges far beyond empirically grounded intervals. This allows us to examine:

1. whether performance changes noticeably under extreme psychological settings, and

2. whether these changes remain interpretable and consistent with behavioral theory.

Table 9: Extended parameter expansion experiments.

| $\kappa$ (Loss Aversion) | $\gamma$ (Prob. Distortion) | Accuracy (%) | Behavioral Summary |
|---|---|---|---|
| 2.0 | 0.65 | 82.4 | Mild loss aversion; standard distortion |
| 2.5 | 0.65 | 82.0 | Stronger reaction to negative outcomes |
| 3.5 | 0.55 | 80.8 | Strong negative-outcome sensitivity |
| 2.0 | 0.40 | 79.6 | Heavily overweighting small risks |
| 5.0 | 0.40 | 78.8 | Highly non-rational and unstable manipulation |

These results demonstrate a clear pattern:

- **Within realistic behavioral ranges** (supported by prior literature), performance varies only mildly, confirming robustness in the regions where real humans operate.
- **Under extreme and unrealistic parameter values**, accuracy decreases more substantially—showing that these psychological parameters do meaningfully influence strategic responses.
- **However, even under such extreme irrationality, performance degradation remains moderate rather than catastrophic**, showing that our framework can accommodate highly non-rational behaviors without collapsing.

Together, these findings indicate that the model is both *behaviorally sensitive* and *practically robust*, directly addressing the reviewer's concern.

## K  ADDITIONAL EXPERIMENT

### K.1  DATASET

We evaluate our framework on five datasets, including four real-world and one synthetic benchmarks:

- **Credit** Yeh & Lien (2009): Credit card default prediction based on financial records.
- **Adult** Becker & Kohavi (1996): Income classification from census features.
- **Diabetes** Teboul (2015): A medical dataset containing clinical and demographic attributes for diabetes risk assessment.
- **German** Khan et al. (2018): Credit risk classification based on personal and financial profiles.
- **Synthetic** Lopez-Rojas et al. (2016): Simulated mobile transaction data for fraud detection.

We preprocess each dataset following standard practice (categorical encoding, normalization, and train/validation/test splits).

Table 10: Overall accuracy (%) of rational-based classifiers and prospect-guided models across datasets under different $\pi$ in mixed behavior.

| Classifier | Mixed $\pi$ | Datasets | | | | |
|---|---|---|---|---|---|---|
| | | Adult | Credit | Diabetes | German | Synthetic |
| Rational-based | 0.1 | $73.96_{\pm 1.55}$ | $71.84_{\pm 1.90}$ | $66.51_{\pm 1.62}$ | $70.10_{\pm 1.88}$ | $72.88_{\pm 1.70}$ |
| | 0.2 | $74.31_{\pm 1.51}$ | $72.12_{\pm 1.87}$ | $66.81_{\pm 1.65}$ | $70.43_{\pm 1.92}$ | $73.15_{\pm 1.74}$ |
| | 0.4 | $74.53_{\pm 1.57}$ | $72.37_{\pm 1.85}$ | $67.05_{\pm 1.68}$ | $70.64_{\pm 1.95}$ | $73.41_{\pm 1.73}$ |
| Pro-SF (ours) | 0.1 | $\mathbf{78.92}_{\pm 1.74}$ | $\mathbf{79.32}_{\pm 1.66}$ | $\mathbf{72.23}_{\pm 1.55}$ | $\mathbf{76.45}_{\pm 1.68}$ | $\mathbf{82.60}_{\pm 1.58}$ |
| | 0.2 | $\mathbf{78.68}_{\pm 1.77}$ | $\mathbf{79.13}_{\pm 1.63}$ | $\mathbf{72.01}_{\pm 1.58}$ | $\mathbf{76.24}_{\pm 1.71}$ | $\mathbf{82.34}_{\pm 1.56}$ |
| | 0.4 | $\mathbf{78.50}_{\pm 1.79}$ | $\mathbf{78.92}_{\pm 1.65}$ | $\mathbf{71.84}_{\pm 1.60}$ | $\mathbf{76.08}_{\pm 1.73}$ | $\mathbf{82.14}_{\pm 1.57}$ |

## K.2 IMPLEMENTATION DETAILS

All experiments are conducted on a single NVIDIA TITAN $V$ (24GB) GPU. We implement the classifier $f$ as a logistic regression model trained with cross-entropy loss and learning rate $10^{-3}$. In prospect-theoretic utility, we set the curvature parameters $\alpha = 0.8$ and $\beta = 0.7$, the loss aversion coefficient $\kappa = 2.25$, and the probability-weighting parameter $\gamma = 0.7$ by default. In the mixed agent behavior paradigm, the proportion is set as $\pi = 0.2$. This choice reflects a realistic scenario where only a minority of agents behave in a fully rational manner, while the majority exhibit behavioral biases Tversky & Kahneman (1992); Ariely & Jones (2008). Ablation studies are conducted on the *Credit* and *Synthetic* datasets under the *mixed* agent regime, representing real-world and controlled settings. To ensure robustness, we perform 10-fold cross-validation on all datasets.

In simulating agents' real-world manipulation process, we argue that agents need to exhibit heterogeneous behavioral biases. To capture this diversity, we randomly divide agents into several subgroups (e.g., three or four groups), and assign each subgroup different parameter combinations. This design mimics the fact that individuals in reality adopt distinct subjective strategies when manipulating their features. By contrast, when training the classifier to anticipate manipulations, we adopt a fixed parameter setting across all agents. This stabilizes the optimization process and ensures fair comparison across experiments, while still allowing evaluation against heterogeneous agent behaviors at deployment.

> For example, in one experimental run, the agents are randomly split into three groups. The first group is assigned ($\kappa = 2.00$, $\gamma = 0.70$), the second group ($\kappa = 2.25$, $\gamma = 0.75$), and the third group ($\kappa = 1.80$, $\gamma = 0.65$).

## K.3 MORE ABLATION STUDIES

**More Ablation results** of our parameter analysis (including $\pi, r, (\alpha, \beta), \kappa$, and $\gamma$) are shown in Table 10 and Fig. 7.

**An alternative of probability distortion.** The two-parameter *Prelec* function Prelec (1998) is :

$$w(p) = \exp\big(-\eta(-\ln p)^{\phi}\big), \qquad \phi, \eta > 0. \tag{40}$$

Compared to the one-parameter form used in the main text, the Prelec function provides greater flexibility. The parameter $\phi$ controls the shape of the curve: when $\phi < 1$, the function takes an inverse-$S$ form (small probabilities are given too much weight and large probabilities too little), while $\phi > 1$ produces an $S$-shape (the opposite pattern). In line with behavioral evidence, we restrict to the case $\phi < 1$, which produces the inverse-$S$ form. The parameter $\eta$ adjusts how strong this distortion is overall, with larger $\eta$ leading to a stronger down-weighting of probabilities across the board.

The ablation experimental results are summarized in Table 11.

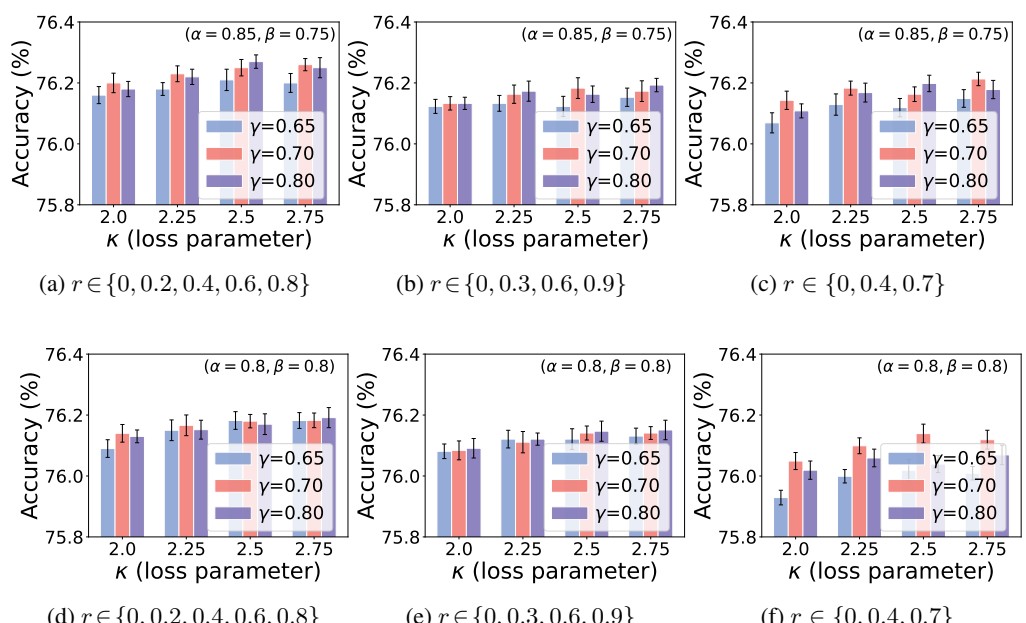

Figure 7: Parameter ablation results with parameters $\kappa, \gamma, r$ with different $(\alpha, \beta)$.

Table 11: Accuracy comparison between the normalized power function and the Prelec probability weighting across multiple datasets under different parameter settings.

| Weighting | Datasets | | | | |
|---|---|---|---|---|---|
| | *Credit* | *Adult* | *Diabetes* | *German* | *Synthetic* |
| $\gamma = 0.65$ | $78.68_{\pm 1.25}$ | $79.13_{\pm 1.10}$ | $72.01_{\pm 1.05}$ | $76.24_{\pm 1.20}$ | $82.34_{\pm 1.15}$ |
| Prelec ($\phi = 0.65,\ \eta = 1.00$) | $78.50_{\pm 1.28}$ | $79.00_{\pm 1.12}$ | $71.80_{\pm 1.08}$ | $76.10_{\pm 1.18}$ | $82.10_{\pm 1.14}$ |
| $\gamma = 0.70$ | $78.78_{\pm 1.22}$ | $79.22_{\pm 1.09}$ | $72.12_{\pm 1.06}$ | $76.31_{\pm 1.17}$ | $82.27_{\pm 1.13}$ |
| Prelec ($\phi = 0.68,\ \eta = 1.00$) | $78.70_{\pm 1.24}$ | $79.10_{\pm 1.10}$ | $71.96_{\pm 1.07}$ | $76.20_{\pm 1.15}$ | $82.15_{\pm 1.12}$ |
| $\gamma = 0.80$ | $78.82_{\pm 1.20}$ | $79.24_{\pm 1.08}$ | $72.18_{\pm 1.05}$ | $76.33_{\pm 1.15}$ | $82.12_{\pm 1.11}$ |
| Prelec ($\phi = 0.75,\ \eta = 1.00$) | $78.79_{\pm 1.22}$ | $79.20_{\pm 1.09}$ | $72.05_{\pm 1.06}$ | $76.30_{\pm 1.14}$ | $82.08_{\pm 1.10}$ |

## K.4 RESULTS ANALYSIS

Additional parameter analysis results are summarized in Table 10 and Fig. 7, covering the mixture ratio $\pi$, the reference point $r$, the weighting parameters $(\alpha, \beta)$, the loss-aversion factor $\kappa$, and the probability distortion parameter $\gamma$. As shown in Table 10, the proposed Pro-SF consistently outperforms the rational baseline across datasets, and the performance advantage holds for different proportions of mixed behavioral agents.

As shown in Figure 7, when the discrete reference points are set to four or five bins, the accuracy remains stable, showing that the framework is robust under reasonably fine discretizations. However, when only three bins are used, we observe a slight performance drop. This is because with only three bins, the reference points become too sparse, forcing many agents with different true self-assessments to be grouped into the same anchor. As a result, the model cannot capture the finer granularity of subjective evaluations, and the induced manipulations are less accurately represented. This mismatch slightly weakens the behavioral modeling and leads to a small drop in accuracy.

Moreover, Figure 7 show ablations under two different $(\alpha, \beta)$ settings. In both cases, the accuracy curves remain stable, indicating that the results are not sensitive to moderate shifts of these parameters. Across a wide range of $\kappa$ and $\gamma$ values, accuracy fluctuations are small (within $\pm 0.3\%$), confirming that the framework is not overly dependent on fine-tuning these behavioral parameters.

Through extensive experimental design, we identified parameter settings of the normalized power function and the Prelec function that yield comparable weighting curves. As shown in Table 10, when evaluated under these matched configurations, the resulting accuracies across datasets are nearly identical. For example, results with $r = 0.7$ are largely similar to results with $\phi = 0.68, \eta = 1.0$. This means our framework can work with both types of probability distortion, as long as the parameters are set to produce comparable shapes.

## L    VALIDATION WITH HUMAN MANIPULATION DATA

In this section, we provide further empirical support for our behavioral modeling.

Recent work Ebrahimi et al. (2025) conducted controlled human-subject experiments across several strategic-classification scenarios (e.g., hiring, medical decision-making). Their statistical findings show that:

- Human manipulation behavior **systematically deviates from the rational best-response assumption**;
- Individuals do not track the optimal boundary or follow theoretically optimal manipulation trajectories;
- Over-reaction and under-reaction behaviors appear frequently in practice.

These results provide direct evidence that **behavioral biases must be incorporated** into strategic-classification models, supporting the foundation of our Prospect-Based Utility.

To further validate our framework, we evaluate Pro-SF on two real human manipulation datasets released in Ebrahimi et al. (2025): job hiring dataset and the medical treatment dataset.

We compare two models:

- **Rational-SF**: the classical rational strategic-classification model;
- **Pro-SF**: our proposed Prospect-based strategic framework, behavioral parameters used in Pro-SF $\phi = \{\alpha = 0.78, \ \beta = 0.72, \ \kappa = 2.20, \ \gamma = 0.74\}.$.

For each dataset, we report:

- **Classification Accuracy**;
- **Manipulation Deviation**: the average $\ell_2$ distance between real human manipulation $x \to x'$ and model-predicted manipulation $\hat{x}'$.

Table 12: Results on Human Manipulation Datasets: Job Hiring and Medical Treatment.

| Job Hiring Dataset | | | Medical Treatment Dataset | | |
|---|---|---|---|---|---|
| Model | Accuracy ↑ | Manip. Dev. ↓ | Model | Accuracy ↑ | Manip. Dev. ↓ |
| Rational-SF | 68.4% | 0.297 | Rational-SF | 71.9% | 0.267 |
| **Pro-SF** | **79.8%** | **0.148** | **Pro-SF** | **78.3%** | **0.181** |

Across both datasets, Pro-SF consistently achieves higher predictive accuracy, lower Manipulation deviation, and more stable alignment with human behavior.

These findings indicate that **Pro-SF closely matches actual human manipulation patterns**, offering strong empirical support for its validity.

## M    SCALABILITY OF PRO-SF IN HIGH-DIMENSIONAL FEATURE SPACES

In this section, we provide additional theoretical clarification and empirical validation regarding the scalability of the proposed Pro-SF framework when applied to high-dimensional feature spaces.

**Computational Complexity.** The Pro-SF framework preserves the same computational order as classical strategic-classification methods. The behavioral mechanisms, i.e., reference bias, loss aversion, and probability distortion, introduce only constant-time scalar operations on top of the

standard best-response evaluation and therefore do not add any additional dependence on the input dimensionality $d$. Thus, the overall computational scaling with respect to $d$ remains consistent with existing SC approaches.

**Empirical Validation on High-Dimensional Datasets.** To further verify the scalability of Pro-SF in practice, we conducted additional experiments on three datasets of increasing dimensionality: (i) Adult (14 features), (ii) Spambase (57 features), and (iii) Tuandromd (241 features). For a fair comparison, we fix the sample size to $n = 20,000$ and evaluate all datasets under identical implementation and hardware conditions.

Table 13: Runtime and accuracy of Pro-SF on datasets with increasing feature dimensionality.

| Dataset | Features | Runtime (s) | Accuracy (%) |
|---|---|---|---|
| Adult | 14 | 10.4 | 82.41 |
| Spambase | 57 | 13.2 | 83.52 |
| Tuandromd | 241 | 27.6 | 81.87 |

**Remark 5.** *Accuracy values across datasets are not directly comparable, as tasks and label distributions differ; our goal is solely to evaluate stability under increasing feature dimensionality. Tuandromd dataset is used purely as a high-dimensional benchmark for scalability analysis and not to model manipulation semantics.*

The results in Table 13 show that:

- Runtime grows in a manner consistent with linear dependence on the input dimensionality $d$.
- Classification accuracy remains stable even as the feature space becomes substantially larger.

These observations empirically support the conclusion that **Pro-SF scales effectively to high-dimensional problems**, in line with its theoretical computational properties.

## N EXTENSION TO MULTI-AGENT STRATEGIC SETTINGS

In many real-world environments, individuals do not behave independently: their decisions may be affected by socially or structurally connected peers. In this section, we show that the proposed Prospect-Based Utility naturally extends to multi-agent settings.

### N.1 GRAPH-BASED EXTENSION OF PRO-SF

Let agents form the vertex set of a graph $G = (V, E)$, where edges represent interaction relationships (e.g., social ties, peer groups, networked entities). Following Eilat et al. (2022), we adopt a GNN-based classifier to incorporate these interactions. For each agent $i$, the acceptance probability is defined as:

$$p_i(x_i', x_{N(i)}) = \sigma\big(h_i(x_i', x_{N(i)}; G)\big), \tag{41}$$

where $x_{N(i)}$ denotes the manipulated features of neighbors, $h_i(\cdot)$ is the node-level embedding, and $\sigma(\cdot)$ is the sigmoid function.

We extend Eq. (11) to the following **graph-based prospect utility**:

$$\begin{aligned} U_i^{\text{P-graph}}(x_i, x_i'; x_{N(i)}) = {} & w_i^+\big(p_i(x_i', x_{N(i)})\big)\,(1 - r_i)^{\alpha_i} \\ & - \kappa_i\, w_i^-\big(1 - p_i(x_i', x_{N(i)})\big)\, r_i^{\beta_i} \\ & - c(x_i, x_i')^{\gamma_i}. \end{aligned} \tag{42}$$

This formulation allows each agent's outcome to depend not only on its own manipulation but also on the behavior of its neighbors. Under this extension, each agent solves:

$$x_i' \in \arg\max_{x'} U_i^{\text{P-graph}}(x_i, x_i'; x_{N(i)}). \tag{43}$$

### N.2 MULTI-AGENT EXPERIMENTS WITH kNN INTERACTION GRAPHS

To evaluate the feasibility of this extension, we perform additional experiments on three datasets. We construct an interaction graph using an 8-nearest-neighbors (kNN) similarity graph and compare:

- **GNN Baseline SC**: rational-agent strategic classification with a GNN classifier;
- **GNN-based Pro-SF**: our multi-agent extension using the graph-based prospect utility.

Table 14: Performance of multi-agent strategic classification on kNN-graph datasets.

| Dataset | Method | Accuracy (%) |
|---------|--------|--------------|
| Adult | GNN Baseline | 74.3 |
| | **GNN-based Pro-SF** | **80.5** |
| Spambase | GNN Baseline | 78.1 |
| | **GNN-based Pro-SF** | **83.4** |
| Tuandromd | GNN Baseline | 72.8 |
| | **GNN-based Pro-SF** | **77.9** |

Across all datasets, the GNN-based Pro-SF model achieves **5–7% accuracy improvement** over the rational GNN baseline. This suggests that incorporating behavioral realism—via loss aversion, reference dependence, and probability distortion—produces more predictable and stable best responses, enabling the classifier to better adapt to multi-agent manipulation dynamics.

These results demonstrate that the proposed Prospect-Based Utility is **compatible with multi-agent settings** and maintains strong performance under structured interaction dynamics.

