# OpenReview forum: "Beyond Rational Illusion: Behaviorally Realistic Strategic Classification"
_ICLR.cc/2026/Conference — Submitted to ICLR 2026_

### Official Review · Reviewer_ZKzn · 2025-10-14

**Soundness:** 2
**Presentation:** 3
**Contribution:** 2
**Rating:** 4
**Confidence:** 3

**Summary:**

This paper extends strategic classification (SC) by incorporating behavioral biases into agents’ decision processes. It introduces the Behaviorally Realistic Strategic Classification (BR-SC) problem and a Prospect-Guided Strategic Framework (Pro-SF) based on prospect theory. The paper models 3 types of behaviors: loss aversion, reference bias, and probability distortion. Experiments on synthetic and real datasets show that Pro-SF remains robust across rational, non-rational, and mixed behavioral patterns.

**Strengths:**

1. The motivation is solid and well-justified. The three behavioral deviations are defined clearly in behavioral economics and represent a realistic extension of strategic classification beyond the rational-agent assumption.

2. The writing is clear and well-structured, with careful explanations of both the theoretical setup and the proposed framework.

3. The experiments are conducted on multiple real and synthetic datasets, showing consistent performance and reasonable robustness under different behavioral settings.

**Weaknesses:**

1. Though I agree with the behavioral deviations from classic SC, I am not sure your modeling of utility is the cleanest way to model irrational SC behaviors. One question is, if we permit different agents to have different utility functions, even the classic norm-based SC cost functions can account for different behavior patterns. For example, if we model the utility as $u(x,x') = f(x') - \beta c(x,x')$ for some agents and $\beta > 1$, it seems that we can also model loss aversion. This makes me feel the settings in this paper are a bit artificial and the contribution might be marginal. Are there any convincing reasons we need to use the **Prospect-Based Utility** Eq. (11)?

2. On lines 358-362, it seems that all parameters for the **Prospect-Based Utility** are set by hand, and there is no "learning" regading these coefficients. Although the experiments include varying the "true" parameters to test robustness, I still feel the authors should discuss choosing parameters from the perspective of the decision-maker.

To conclude, my main concern is that the utility function is too artificial, but it seems that the key contribution lies in this modeling of utility.

**Questions:**

Could you provide more context justifying your modeling?

---

> ### Author Response · Authors · 2025-11-18
> **We would like to supplement the clarification and experimental results to address your concern (weakness 1)**
>
> Dear Reviewer,
>
> Thank you for your thoughtful comments. We would like to address your concerns **point by point**.
>
> > **In weakness 1:** One question is, if we permit different agents to have different utility functions, even the classic norm-based SC cost functions can account for different behavior patterns.
>
> **Response:**
>
> We sincerely thank the reviewer for raising this insightful point.
>
> - We agree that assigning different utility functions (e.g., different $\beta$) to different agents can create **behavioral diversity**.
>
>   - However, such diversity still lies within the rational framework and mainly **reflects how different agents behave under rational optimization**.
>
> - In practice, agents’ behaviors can be divided into **rational** and **non-rational** behaviors, which form **two distinct subsets** of the overall behavior space.
>   - We add a picture to illustrate this relationship in our revised paper (Figure 6 in Appendix H, lines 1119).
>
> - We also appreciate the reviewer's example $u(x,x′) = f(x') - \beta c(x,x')$.
>   - Increasing  $\beta$ indeed makes agents more cost-sensitive, but its behavior remains confined within the **rational behavior subset**.
>
> - We would like to clarify that human *loss aversion* is not a matter of merely scaling up cost. It arises from **structural psychological asymmetries**, that cannot be captured by simply scaling $\beta$.
>
> - Therefore, the core purpose of Prospect-based Utility is **not** to artificially complicate the model, but to capture the **non-rational behaviors that rational utility models cannot express**, thereby aligning more closely with how agents actually behave in real scenarios.
>
> - **To further address this concern, we add a new experiment as follows:**
>
>   - We construct a more realistic environment in which: (i) **50%** of agents behave **rationally**, and (ii) **50%** of agents behave **non-rationally**.
>
>   - In this mixed environment, we train two classifiers:
>
>     - **Classifier trained with $U_\beta$**: anticipating agent manipulation using $u(x,x') = f(x') - \beta c(x,x')$.
>     - **Classifier trained with $U_P$**: anticipating manipulation using our Prospect-based Utility.
>
>   - Then we evaluate both classifiers **under the same mixed environment**, and report the results:
>
>     **Table 1. Performance of classifiers with different agent modeling.**
>
>     | Classifier          | Model Parameters      | Post-gaming Accuracy |
>     | ---- | ------------- | -------------------- |
>     | f with $U_P$ (ours) | $\alpha=0.80, \beta=0.5, \kappa=2.20, \gamma=0.7$ | 81.6%     |
>     | f wth $U_{\beta}$   | $\beta = 1.5$   | 79.1%   |
>     | f wth $U_{\beta}$   | $\beta = 1.7$   | 79.3%   |
>     | f wth $U_{\beta}$   | $\beta = 1.9$    | 79.1%   |
>     | f wth $U_{\beta}$   | $\beta = 2.0$     | 78.9%    |
>     |  |  |
>
> - Therefore, we would like to clarify that Prospect-based Utility is not merely a stylistic choice, but a **necessary and structurally richer model** for capturing realistic strategic behaviors.
>
>
> **Paper Revision.** We have **revised** our paper and supplemented this part in Appendix H (Lines 1118-1146).
>
> -----
>
> >  **In Weakness 1:** Are there any convincing reasons we need to use the **Prospect-Based Utility** Eq. (11)?
>
> **Response:**
>
> - We fully agree that the classic SC framework can generate **heterogeneous behaviors**. However, it still assumes that all agents behave optimally in rational behavior, which **cannot account for some systematic behaviors observed in real settings**, including:
>   - agents who **do not react until crossing a psychological threshold**, even though marginal gains have already exceeded marginal costs;
>   - agents who **forgo manipulation even when doing so is clearly profitable** (under-defense);
>   - agents who **over-adjust beyond the rational optimum**, incurring unnecessary cost (over-defense);
>
> - To overcome these realistic gaps, Prospect-Based Utility (Eq. 11) is proposed as a **structured and theoretically supported extension** for the SC framework.
>
>   - This formulation integrates three key mechanisms, i.e., **loss aversion**, **probability distortion**, and **reference dependence**, in the cleanest possible form.
>
>   - Each of these mechanisms has been well established in **Prospect Theory** and later validated across behavioral economics and psychology, providing a principled foundation for modeling non-rational behavior.
>
> - The classical SC framework remains theoretically valuable. Our goal is not to replace it, but to **extend it in a behaviorally grounded way** that better reflects how humans actually make strategic decisions.
>
> - We would like to **clarify that the contribution of this work** lies not only in proposing a new utility formulation, but also in highlighting behavioral patterns that rational models may overlook and offering a structured, testable framework to better capture them.
>
>
> **Paper Revision.** We have **revised** our paper and clarified this motivation in Section 4.1 (Lines 234-238).

---

> ### Author Response · Authors · 2025-11-18
> **We would like to supplement some clarification and experimental results to address your concern (weakness 2)**
>
> > It seems that all parameters for the **Prospect-Based Utility** are set by hand, and there is no "learning" regarding these coefficients ... should discuss choosing parameters from the perspective of the decision-maker.
>
> **Response:**
>
> We **agree that parameterization should be discussed from the decision-maker’s perspective**. In response, we supplement corresponding **theoretical justification and experimental validation**.
>
> - We denote the parameters of the Prospect-Based Utility (Eq. 11) as $\theta=(α, β, κ, γ).$
> - To infer these parameters from observed manipulation (x,x'), we adopt a likelihood-based inverse optimization objective:
>   $$
>   \theta^{*}=\arg\max_{\theta} \sum_i\log P_{\theta}(x_i' \mid x_i).
>   $$
>   - This objective corresponds to an **inverse optimal decision problem**, which can be addressed using standard tools from inverse reinforcement learning and differentiable inverse optimization. We follow the prior work [1,2,3].
> - To evaluate whether this optimization can reliably recover behavioral parameters, we **conduct the following experiment**:
>
>   **(1) Parameter learning on real human data.**
>   - We evaluate the above parameter-learning procedure using the two real human manipulation datasets provided in prior work[4]: **job hiring** data and **medical treatment** data.
>   - We infer the behavioral parameters $\hat{\theta}$ and evaluate the Pro-SF classifier $\hat{\theta}$.
>
>     **Table 2. Learned parameters and accuracy on real human data.**
>
>     |Dataset|Parameters $\hat{\theta}$| Accuracy(%)|
>     |-|-|-|
>     |Hiring|{α=0.79, β=0.73, κ=2.18, γ=0.72} |79.8%|
>     |Medical|{α=0.81, β=0.70, κ=2.25, γ=0.71} |78.3%|
>     |
>
>   - These results demonstrate that **Prospect-based behavioral parameters can be reliably inferred from real manipulation data.**
>
>   **(2) Parameter learning on synthesis datasets.**
>   - We simulate a **mixed and heterogeneous population**, where each agent is randomly assigned to one of two groups: rational and non-rational.
>   - Specifically, we vary **the proportion $\pi$ of rational agents in $\pi \in {0.2,0.4,0.6}$**. For example, if $\pi = 0.2$, we set:
>
>     (1) **80% non-rational agents**, following the Prospect-Based Utility;
>     - For each behavioral agent, its parameters 𝜃 are drawn from a uniform distribution defined over the parameter ranges established in prior behavioral-economics literature.
>     - Following, the manipulated feature is generated as a **noisy prospect-guided decision**:
>     $$
>     x'_i=\arg \max U_p(x_i, x'_i; \theta_i)+\epsilon_i, \quad \epsilon_i \sim N(0, \sigma^2)
>     $$
>
>      (2) **20% fully rational agents** follow the standard strategic-classification utility,  i.e., maximizing the acceptance probability minus the manipulation cost.
>   - After obtaining the manipulated feature pairs (x,x'), we **estimate the behavioral parameters $\hat{\theta}$ using the inverse optimal decision framework** introduced above.
>   - We report the learned $\hat{\theta}$ and evaluate its test accuracy on each dataset:
>
>     **Table 3. Learned parameters and accuracy with $\pi = 0.2$.**
>
>     |Dataset|Parameters $\hat{\theta}$| Accuracy (%)|
>     |-|-|-|
>     |Adult|{α=0.78, β=0.72, κ=2.20, γ=0.74}|81.42|
>     |Credit|{α=0.80, β=0.70, κ=2.15, γ=0.71}|82.30|
>     |German|{α=0.81, β=0.70, κ=2.25, γ=0.72} | 79.21|
>     |
>
>     **Table 4. Learned parameters and accuracy with  $\pi = 0.4$.**
>
>     |Dataset|Parameters $\hat{\theta}$|Accuracy (%) |
>     |-|-|-|
>     |Adult|{α=0.78, β=0.74, κ=2.20, γ=0.74}|81.32|
>     |Credit|{α=0.80, β=0.70, κ=2.10, γ=0.70}|82.20|
>     |German|{α=0.80, β=0.72, κ=2.18, γ=0.71}|79.25 |
>     |
>
>     **Table 5. Learned parameters and accuracy with $\pi = 0.6$.**
>
>     |Dataset|Parameters $\hat{\theta}$|Accuracy (%)|
>     |-|-|-|
>     |Adult|{α=0.78, β=0.76, κ=2.05, γ=0.70} |81.28|
>     |Credit|{α=0.78, β=0.74, κ=2.03, γ=0.68}|82.16|
>     |German|{α=0.80, β=0.73, κ=2.10, γ=0.69}|79.08|
>     |
> - This demonstrates that:
>   - **behavioral parameters are learnable from heterogeneous strategic behavior**.
>   - **Pro-SF is robust to parameter estimation and does not rely on knowing parameters**.
>
> **Paper Revision.** We have revised our paper in **Section 5.3 (lines 409-418) and supplemented detailed experimental results and analysis in Appendix I (lines 1149-1234)**.
>
> ---
> *Reference*
>
> [1] Ng, Andrew Y., and Stuart Russell. "Algorithms for inverse reinforcement learning." *Icml*. Vol. 1. No. 2. 2000.
>
> [2] Amos, Brandon, and J. Zico Kolter. "Optnet: Differentiable optimization as a layer in neural networks." International conference on machine learning. PMLR, 2017.
>
> [3] Agrawal, Akshay, et al. "Differentiable convex optimization layers." Advances in neural information processing systems 32 (2019).
>
> [4] Ebrahimi, Raman, Kristen Vaccaro, and Parinaz Naghizadeh. "The double-edged sword of behavioral responses in strategic classification: Theory and user studies." *Proceedings of the 2025 ACM Conference on Fairness, Accountability, and Transparency*. 2025.

---

> ### Author Response · Authors · 2025-11-18
> **We would like to supplement two specific contexts to address your concern (quesiont and the conclusion)**
>
> > **Question:** Could you provide more context justifying your modeling?
>
> **Response:**
>
> We thank the reviewer for raising this important question.
>
> - Before giving concrete examples, we briefly review the **three core behavioral mechanisms** our Prospect-Based Utility is designed to capture:
>   - **Reference bias**: individuals evaluate outcomes relative to a personal baseline rather than an absolute value.
>   - **Loss aversion**: losses loom larger than equivalent gains.
>   - **Probability distortion**: individuals overweight small probabilities and underweight large ones.
>
> Next, we would like to **provide two specific contexts** to support our modeling.
>
> **1. Financial Investment**
>
> - Consider **two investors facing the same investment opportunity**: invest \$ 10,000 with a 60% chance to earn \$ 20,000 and a 40% chance to lose all principal.
>
>   - **Investor A** has \$100,000 in savings,
>
>   - **Investor B** has only \$20,000.
>
>
> - Under the **classic SC framework**, because the expected gain exceeds the monetary cost, the model predicts that *both* A and B should take the action.
> - However, real behavior differs substantially:
>
>   - Investor A is inclined to invest, whereas Investor B often chooses not to act.
>   - This is because their **savings place them at different reference points**.
>     - Losing \$ 10,000 corresponds to only **10%** of  Investor A total wealth but **50%** of Investor B.
>     - Additionally, loss aversion also makes **the same monetary loss psychologically far more severe** for B.
> - Therefore, we propose Prospect-Based Utility to capture this behavior.
>
>   - It assigns each agent’s **individual reference point r** from their state and incorporates **asymmetric loss weighting $\kappa$** to reflect psychological sensitivity to losses.
>   - As a result, the same action incurs a larger *effective psychological cost* for Investor B, leading A and B to choose different optimal manipulation strategies.
>
> **2.Disease Screening:**
>
> - Considering a disease-screening scenario, even when the true prevalence of a disease is extremely low (e.g., 0.1%), individuals may repeatedly undergo testing and stockpile medication.
> - This pattern reflects a form of *psychologically driven non-rational behavior*：
>
>   - Individuals overweight very small probabilities, treating a 0.1% risk as something that “cannot be ignored”.
>
>   - After an alarm or a stressful event, the psychological reference point may shift from *“I am healthy”* to *“I might already be at risk”*, sharply increasing the perceived cost of not taking action.
>
> - So the classical SC model predicts that a 0.1% risk should induce only minimal adjustment—not repeated testing or persistent submission of extra documents.
>
>   - In other words, classic SC **cannot capture this type of over-acting behavior**.
>
> - **Prospect-Based Utility** naturally captures this phenomenon through:
>
>   - a **probability weighting function w(p)** that amplifies small perceived risks, and
>
>   - a **reference-point term r** that models shifts in psychological baselines across different agents and contexts.
>
> **Paper Revision.** We have **revised** our paper and supplemented this detailed context in Appendix G.1 and G.2 (lines 1087-1113).
>
> ------
>
> **We appreciate your clear summary of the concerns:**
> > my main concern is that the utility function is too artificial, but it seems that the key contribution lies in this modeling of utility.
>
> **Response:**
>
> - To summarize, our proposed utility function is **not intended as an artificial design**, but is designed to capture **objective behavioral patterns observed in real scenarios**.
>
> - The three mechanisms we adopt, i.e., **reference bias, loss aversion, and probability distortion**, are established in Prospect Theory and other prior studies in behavioral economics.
>
> - Moreover, our additional experiments show that **the parameters of this behavioral model are learnable from strategic manipulation data**, rather than being manually chosen.
>
> - Overall, our contributions are twofold:
>   - We highlight the realistic behavior of agents that classical SC cannot capture, motivating a connection between behavioral economics and strategic classification.
>   - We develop a practicable, structured, and learnable framework that models realistic agent behavior.
>
> - We summarize all revisions made to the paper in the table below for your convenience.
>
> **Table 6. Revision Summary**
> |Comments| Location of Revisions in Our Revised Paper|
> |-----|---------|
> | Weakness 1 | We revised **Section 4.1 (lines 234–238), Section 4.3 (lines 348-350), and  Appendix H (lines 1118–1146)**. |
> | Weakness 2 | We revised **Section 5.3 (lines 409–418) and Appendix I (lines 1149-1234)**. |
> | Question 1 | We revised **Appendix G.1 and G.2 (lines 1087–1113).**       |
> || |
>
>
> Thank you for your thoughtful comments. If you have **any further concerns or questions, please feel free to raise them**. We would be glad to continue the discussion and clarify them in more detail.

---

> > ### Comment · Reviewer_ZKzn · 2025-11-20
> >
> > Thanks for the extensive rebuttal. I totally understand that the behavior patterns you proposed in the paper can be more versatile in explaining certain psychological patterns in real-world SC problems. I think it's fair to regard this as a contribution. My concern remains that, since the classic model can indeed incorporate the settings presented in this paper, the contribution might be somewhat marginal. Inferring the parameters is beneficial and enhances the paper's strength.
> >
> > Overall, I do not mind accepting the paper and I increase my rating to 6.

---

> > > ### Author Response · Authors · 2025-11-20
> > >
> > > Dear Reviewer,
> > >
> > > Thank you for your supportive feedback.
> > >
> > > We sincerely appreciate your insightful suggestions, which help us improve this work and better articulate our contributions.

---

### Official Review · Reviewer_Y5cB · 2025-10-25

**Soundness:** 2
**Presentation:** 3
**Contribution:** 2
**Rating:** 2
**Confidence:** 4

**Summary:**

This work considers the strategic classification setting, where a principal reveals a classifier and the user can then manipulate the data they submit to the classifier. The classic Stackelberg model considers rational users who perfectly best responds to the classifier - the principal based their strategic optimum based on such an assumption of rationality about the users. Inspired by behavioural econ, this paper looks to challenge that by introducing 3 behavioral tendencies: loss aversion, reference bias, probability distortion. They propose a new behavioural model of user decision making and empirically consider a Stackelberg optimum against such a model.

I think the problem is very well motivated and the three behavioral tendencies have long justification in economics and psycology. Beyond an elaboration of the model, however, I struggle to find the major contribution of the work. The theoretical results on under and over defense are immediate. Given the proposed user behvaioural model, no theoretical results (on learnability or optimizability of the stackelberg equilibrium) are given. I do think it's a major shortcoming, if compelling empirical work was provided. But this falls short on several grounds. First, behavioral models should be backed with some actual human evaluations - they should be communicated the classifier and assessed how they actually manipulate here. Otherwise, one naturally expects the behaviour aware classifier to outperform the control when all agents are assumed to behave exactly as the former expects. Second, are the principals supposed to know the behavioural parameters like $\kappa, r$? Why would we expect this to be the same for all users. These are tricky questions, but one where we could have more results on if real field deployed experiments were conducted.

As such, my rating is based on the inconclusive nature of both the theoretical and empirical results.

**Strengths:**

See above.

**Weaknesses:**

See above

**Questions:**

See above.

---

> ### Author Response · Authors · 2025-11-20
> **We would like to supplement the clarification and experimental results to address your concerns（1）**
>
> Dear Reviewer,
>
> Thank you for your thoughtful comments. We would like to **address your concerns point by point**.
>
> > Given the proposed user behvaioural model, no theoretical results (on learnability or optimizability of the stackelberg equilibrium) are given.
>
> **Response**
>
> Thank you for raising this insightful point. To address your concern about the theoretical results of Pro-SF, we **add a theoretical analysis of the existence of a Stackelberg equilibrium of the Pro-SC**.
>
> - **To establish such an equilibrium, our analysis proceeds by verifying two key components:**
>   (1) the agent’s prospect-guided best-response $b^* (x)$ exists (the follower);
>   (2) the leader’s optimal classifier $f^* ()$ exists (the leader).
>
> - First, from the agent’s perspective:
>
>   - Let $\theta$ denote the parameter set of the classifier $f$. For any classifier $f_{\theta}$, the agent searches over the feasible feature space $\mathcal{X}$ for an optimal manipulation:
>
>   $$
>   b^*(x) \in \arg \max_{x'}U_P(x,x′;\theta).
>   $$
>
>   - In practice, the feature space $\mathcal{X}$ is bounded and closed, since individuals’ manipulable features (e.g., income range, test score range, health indicators) naturally have realistic upper and lower limits.
>   - Moreover, the Pro-SF utility the Pro-SF utility $U_P(x,x′;\theta)$ is continuous in x′.
>   - Therefore, the agent’s optimization problem satisfies Weierstrass extreme value theorem[1,2]: **a continuous function defined over a bounded closed set must attain at least one maximizer**.
>
> - From the classifier’s perspective:
>
>   - We **assume the parameter space $\Theta$ is bounded and closed**, which is **standard in practice explicit parameter constraints**.
>
>   - Given the the prospect-guided best response $b^* (x; \theta)$, **The classifier’s objective can be written as:**
>     $$
>     F(θ)=\mathbf{E}[\mathcal{L}(f_{\theta}(b^*(x;\theta)),y)].
>     $$
>
>   - For common models (e.g., linear or logistic), both the prediction $f_\theta(\cdot)$ and the loss $\mathcal{L}$ are **continuous** in $\theta$.
>
>   - Therefore, according to Weierstrass condition[1,2], the an optimal classifier parameter also exists.
>
> - Combining the above, **under the Prospect-Based Utility**, the strategic classification game continues to **admit a Stackelberg equilibrium.**
>
> - To complement this argument, we also conducted a simple iterative experiment.
>
>   - We record the classification accuracy and the average manipulation magnitude after the interaction between agents and the classifier.
>
>        **Table 1. Performance of Pro-SF with Iteration t**
>
>     | Iteration t | Accuracy ↑ | Avg. Manipulation ‖x^{(t)} – x^{(t-1)}‖₂ ↓ |
>     | ----------- | ---------: | -----------------------------------------: |
>     | 1           |      81.2% |                                       0.31 |
>     | 5           |      81.7% |                                       0.08 |
>     | 10          |      82.1% |                                       0.04 |
>     | 15          |      82.2% |                                       0.02 |
>     | 20          |      82.2% |                                       0.01 |
>     |||
>
> - We observe that (i) accuracy improves only marginally after a few iterations, and (ii) the per-step manipulation shrinks towards zero.
>
>   - This is consistent with **Pro-SF driving the system toward an equilibrium point where agents have almost no further incentive to manipulate**.
>
>
> **Paper Revision.** We have supplemented this analysis on the Stackelberg equilibrium in **Appendix F (pages 20-21)** of our revised paper.
>
>
> ----
> *Reference*
>
> [1] Boyd, Stephen, and Lieven Vandenberghe. *Convex optimization*. Cambridge university press, 2004.
>
> [2] Stone, Marshall H. "The generalized Weierstrass approximation theorem." *Mathematics Magazine* 21.5 (1948): 237-254.

---

> ### Author Response · Authors · 2025-11-20
> **We would like to supplement the clarification and experimental results to address your concerns (2)**
>
> > behavioral models should be backed with some actual human evaluations - they should be communicated to the classifier and assessed how they actually manipulate here ... behave exactly as the former expects.
>
> **Response:**
>
> We sincerely thank the reviewer for the valuable comments.
>
> To address your concern, we would like to **clarify the foundations of our behavioral model from the following two perspectives**.
>
> - **First, we would like to provide real-world human studies to support our behavioral model.**
>
>   - In the prior work [1], the authors conducted user studies across different scenarios (e.g., hiring, medical treatment), specifically designed for *strategic classification*.
>   - Their statistical results indicate that **human manipulation behavior deviates substantially from the rational best-response assumption**, as individuals do not follow theoretically optimal manipulation trajectories.
>   - These findings offer direct human-based evidence for incorporating *behavioral biases* into strategic classification models.
>
> - **Second, we supplement a real manipulation experiment.**
>
>   - We compare two models:
>
>     - **Baseline 1**: the classical *rational* SC model.
>     - **Baseline 2**: our proposed behavioral model **Pro-SF**.
>
>   - We evaluate both models on two sets of real human manipulation datasets provided in prior work[1]:
>
>     - Job Hiring Data.
>     - Medical treatment Data.
>
>   - We report two metrics for each dataset:
>
>     - Accuracy.
>     - Manipulation deviation: the average L2 distance between the real human manipulation $x \rightarrow x'$ and the model-predicted manipulation$′\hat{x}'$
>
>     **Table 2. Results on Job Hiring Data**
>
>     | Model       | Accuracy ↑ | Manipulation deviation ↓ |
>     | ----------- | ---------- | ------------------------ |
>     | Rational-SF | 68.4%      | 0.297                    |
>     | Pro-SF      | **79.8%**  | **0.148**                |
>     |             |            |                          |
>
>     **Table 3. Results on Medical Treatment Data**
>
>     | Model       | Accuracy ↑ | Manipulation deviation ↓ |
>     | ----------- | ---------- | ------------------------ |
>     | Rational-SF | 71.9%      | 0.267                    |
>     | Pro-SF      | **78.3%**  | **0.181**                |
>     |
>
>
>
>   - Across both datasets, Pro-SF consistently achieves higher predictive accuracy, lower Manipulation deviation.
>   - These results indicate that Pro-SF demonstrates stable advantages **when evaluated directly on real human manipulation data**, supporting its practical validity.
>
> - **Note:**
>
>   - For clarity, **the behavioral parameters used in Pro-SF are set consistently ** $\theta = \{\alpha = 0.8,\; \beta = 0.72,\; \kappa = 2.20,\; \gamma = 0.72\}.$
>
>   - These values are inferred from the real human manipulation pairs $(x, x')$ through a likelihood-based estimation procedure. Specifically, we solve:
>     $$
>     \theta^{*} = \arg\max_{\theta} \sum_i \log P_{\theta}(x_i' \mid x_i)
>     $$
>
>   - Due to space constraints, we provide a full explanation of this estimation process in the next part of our response.
>
> **Paper Revision.** We have supplement this in **Appendix L (lines 490-494)** of our revised paper.
>
> -----
>
> *Reference*
>
> [1] Ebrahimi, Raman, Kristen Vaccaro, and Parinaz Naghizadeh. "The double-edged sword of behavioral responses in strategic classification: Theory and user studies." Proceedings of the 2025 ACM Conference on Fairness, Accountability, and Transparency. 2025.

---

> ### Author Response · Authors · 2025-11-20
> **We would like to supplement the clarification and experimental results to address your concerns（3）**
>
> > Are the principals supposed to know the behavioural parameters like κ, γ? Why would we expect this to be the same for all users.
>
> **Response:**
>
> We **agree that parameterization should be discussed from the decision-maker’s perspective**. In response, we supplement corresponding **theoretical justification and experimental validation**.
>
> - We denote the parameters of the Prospect-Based Utility (Eq. 11) as $\theta=(α, β, κ, γ).$
> - To infer these parameters from observed manipulation (x,x'), we adopt a likelihood-based inverse optimization objective:
>   $$
>   \theta^{*}=\arg\max_{\theta} \sum_i\log P_{\theta}(x_i' \mid x_i).
>   $$
>   - This objective corresponds to an **inverse optimal decision problem**, which can be addressed using standard tools from inverse reinforcement learning and differentiable inverse optimization. We follow the prior work [1,2,3].
> - To evaluate whether this optimization can reliably recover behavioral parameters, we **conduct the following experiment**:
>
>   **(1) Parameter learning on real human data.**
>   - We evaluate the above parameter-learning procedure using the two real human manipulation datasets provided in prior work[4]: **job hiring** data and **medical treatment** data.
>   - We infer the behavioral parameters $\hat{\theta}$ and evaluate the Pro-SF classifier $\hat{\theta}$.
>
>     **Table 4. Learned parameters and accuracy on real human data.**
>
>     |Dataset|Parameters $\hat{\theta}$| Accuracy(%)|
>     |-|-|-|
>     |Hiring|{α=0.79, β=0.73, κ=2.18, γ=0.72} |79.8%|
>     |Medical|{α=0.81, β=0.70, κ=2.25, γ=0.71} |78.3%|
>     |
>
>   - These results demonstrate that **Prospect-based behavioral parameters can be reliably inferred from real manipulation data.**
>
>   **(2) Parameter learning on synthesis datasets.**
>   - We simulate a **mixed and heterogeneous population**, where each agent is randomly assigned to one of two groups: rational and non-rational.
>   - Specifically, we vary **the proportion $\pi$ of rational agents in $\pi \in {0.2,0.4,0.6}$**. For example, if $\pi = 0.2$, we set:
>
>     (1) **80% non-rational agents**, following the Prospect-Based Utility;
>     - For each behavioral agent, its parameters 𝜃 are drawn from a uniform distribution defined over the parameter ranges established in prior behavioral-economics literature.
>     - Following, the manipulated feature is generated as a **noisy prospect-guided decision**:
>     $$
>     x'_i=\arg \max U_p(x_i, x'_i; \theta_i)+\epsilon_i, \quad \epsilon_i \sim N(0, \sigma^2)
>     $$
>
>      (2) **20% fully rational agents** follow the standard strategic-classification utility,  i.e., maximizing the acceptance probability minus the manipulation cost.
>   - After obtaining the manipulated feature pairs (x,x'), we **estimate the behavioral parameters $\hat{\theta}$ using the inverse optimal decision framework** introduced above.
>   - We report the learned $\hat{\theta}$ and evaluate its test accuracy on each dataset:
>
>     **Table 5. Learned parameters and accuracy with $\pi = 0.2$.**
>
>     |Dataset|Parameters $\hat{\theta}$| Accuracy (%)|
>     |-|-|-|
>     |Adult|{α=0.78, β=0.72, κ=2.20, γ=0.74}|81.42|
>     |Credit|{α=0.80, β=0.70, κ=2.15, γ=0.71}|82.30|
>     |German|{α=0.81, β=0.70, κ=2.25, γ=0.72} | 79.21|
>     |
>
>     **Table 6. Learned parameters and accuracy with  $\pi = 0.4$.**
>
>     |Dataset|Parameters $\hat{\theta}$|Accuracy (%) |
>     |-|-|-|
>     |Adult|{α=0.78, β=0.74, κ=2.20, γ=0.74}|81.32|
>     |Credit|{α=0.80, β=0.70, κ=2.10, γ=0.70}|82.20|
>     |German|{α=0.80, β=0.72, κ=2.18, γ=0.71}|79.25 |
>     |
>
>     **Table 7. Learned parameters and accuracy with $\pi = 0.6$.**
>
>     |Dataset|Parameters $\hat{\theta}$|Accuracy (%)|
>     |-|-|-|
>     |Adult|{α=0.78, β=0.76, κ=2.05, γ=0.70} |81.28|
>     |Credit|{α=0.78, β=0.74, κ=2.03, γ=0.68}|82.16|
>     |German|{α=0.80, β=0.73, κ=2.10, γ=0.69}|79.08|
>     |
> - This demonstrates that:
>   - **behavioral parameters are learnable from heterogeneous strategic behavior**.
>   - **Pro-SF is robust to parameter estimation and does not rely on knowing parameters**.
>
> **Paper Revision.** We have revised our paper in **Section 5.3 (lines 409-418) and supplemented detailed experimental results and analysis in Appendix I (lines 1149-1234)**.
>
> ---
> *Reference*
>
> [1] Ng, Andrew Y., and Stuart Russell. "Algorithms for inverse reinforcement learning." *Icml*. Vol. 1. No. 2. 2000.
>
> [2] Amos, Brandon, and J. Zico Kolter. "Optnet: Differentiable optimization as a layer in neural networks." International conference on machine learning. PMLR, 2017.
>
> [3] Agrawal, Akshay, et al. "Differentiable convex optimization layers." Advances in neural information processing systems 32 (2019).
>
> [4] Ebrahimi, Raman, Kristen Vaccaro, and Parinaz Naghizadeh. "The double-edged sword of behavioral responses in strategic classification: Theory and user studies." *Proceedings of the 2025 ACM Conference on Fairness, Accountability, and Transparency*. 2025.

---

> ### Author Response · Authors · 2025-11-20
> **We would like to supplement a summary table to outline the revisions to our manuscript.**
>
> For your convenience, we have included **a summary table detailing the changes in the revised version** of our paper.
>
> **Table 8. Revision Summary**
>
>
> | Comments   | Location of Revisions in Our Revised Paper                   |
> | ---------- | ------------------------------------------------------------ |
> | Concern 1 | We revised **Appendix F (pages 20-21)**. |
> | Concern 2 | We revised **Appendix L (lines 490-494)**. |
> | Concern 3 | We revised **Section 5.3 (lines 409–418) and Appendix  I (lines 1149-1234).**       |
> |||
>
> Thank you for your thoughtful comments.
>
> **If you have any further concerns or questions, please feel free to raise them.** We would be glad to continue the discussion and clarify them in more detail.

---

> ### Author Response · Authors · 2025-11-26
>
> Dear Reviewer Y5cB,
>
> We sincerely appreciate your valuable comments.
>
> If any point could benefit from further clarification, we would be truly grateful if you could let us know. We welcome the opportunity to discuss them and provide any additional information needed.
>
> Many thanks,
>
> The authors of #2185

---

> ### Author Response · Authors · 2025-11-28
> **We would like to supplement a summary of our theoretical contribution for better addressing your concerns**
>
> Dear Reviewer Y5cB,
>
> To better address your concerns, we provide a **brief clarification regarding the theoretical contribution** of our paper.
>
> - **First**, our paper provides the **first theoretical characterization of two systematic failures** of the classical rationality assumption in strategic classification, i.e., **over-defense** and **under-defense**.
>
>   - These phenomena have not been formally identified or explained in the existing SC literature. Our analysis shows that these failures arise structurally from behavioral biases rather than from noise or optimization artifacts.
>   - The precise conditions and full proofs are provided in the appendix, **representing a substantive theoretical addition to current SC**.
>
> - **Second**, the **Prospect-Guided Utility** is not a replacement for the rational model, but rather its **strict superset**.
>
>   - The framework naturally recovers the classical SC formulation as a special case, while also capturing behavioral agents exhibiting loss aversion, reference bias, and probability distortion.
>   - This unified framework makes it **possible to analyze, within a single model, how different behavioral patterns alter agents’ best responses and the resulting optimal classifier**.
>
> - **Third, following your thoughtful suggestions,** we have supplemented a formal analysis of the **Stackelberg equilibrium** under the Prospect Strategic Framework and established its **learnability and solvability guarantees**.
>
>   - We further support these results with experiments on real human manipulation data, confirming that the learned behavioral parameters align well with observed human behaviors.
>
> - **Fourth, from a broader social science perspective,** our work establishes a meaningful bridge between behavioral economics and strategic machine learning.
>
>   - **A wide range of social science applications are increasingly converging with machine learning and economics**, such as auction theory[1,2], social welfare optimization[3,4], congestion externality[5], and incentive-compatible design [6].
>   - Our work provides **a foundational contribution** by integrating realistic behavioral modeling with game-theoretic learning frameworks.
>   - We hope this foundation will **facilitate future interdisciplinary research and applications of behavioral models** in societal and policy-related decision-making contexts.
>
>
>
>   -----
>
>   Reference
>
>
>
>   [1] Duetting, Paul, Vahab Mirrokni, Renato Paes Leme, Haifeng Xu, and Song Zuo. "Mechanism design for large language models." In *Proceedings of the ACM Web Conference 2024*, pp. 144-155. 2024.
>
>   [2] Prasad, Siddharth, Maria-Florina F. Balcan, and Tuomas Sandholm. "Bicriteria multidimensional mechanism design with side information." *Advances in Neural Information Processing Systems* 36 (2023): 40832-40852.
>
>   [3]Haghtalab, Nika et al. “Maximizing Welfare with Incentive-Aware Evaluation Mechanisms.” *International Joint Conference on Artificial Intelligence* (2020).
>
>   [4] Heidari, Hoda, et al. "Fairness behind a veil of ignorance: A welfare analysis for automated decision making." *Advances in neural information processing systems* 31 (2018).
>
>   [5] Hossain, Safwan, Evi Micha, Yiling Chen, and Ariel D. Procaccia. "Strategic Classification With Externalities." In *The Thirteenth International Conference on Learning Representations*.
>
>   [6] Yu, Guanghui, et al. "Encoding human behavior in information design through deep learning." *Advances in neural information processing systems* 36 (2023): 7506-7528.

---

### Official Review · Reviewer_kBEE · 2025-11-01

**Soundness:** 2
**Presentation:** 3
**Contribution:** 2
**Rating:** 6
**Confidence:** 2

**Summary:**

This paper studied the case where full rationality assumption is broken in strategic classification (SC)  and proposed the Behaviorally Realistic Strategic Classification (BR-SC) framework. It also introduced the Prospect-Guided Strategic Framework (Pro-SF) to deal with the problem. Pro-SF explicitly models loss aversion, reference bias, and probability distortion, extending the standard Stackelberg game formulation of SC.

**Strengths:**

- Formulate the BR-SC problem, model three key deviations from rationality, focusing on loss aversion, probability distortion, and reference bias. The over-defense and under-defense case and the corresponding accuracy degradation is introduced.
- The paper is well-organized, and related literature is discussed.
- Experiments on synthetic and real-world datasets show that Pro-SF outperforms rational-based baselines under mixed or non-rational agent behaviors.

**Weaknesses:**

- The strategic behaviour and limited rationality is not new in the literature, and the combination of two is also studied a lot.
- For all three cases: loss aversion, probability distortion, and reference bias, the psychological parameters $(\alpha, \beta, \kappa, \gamma, K)$ are selected heuristically. There is no empirical evidence that they correspond to actual human behavior.
- In the experiment, the sensitivity test only provides a small deviation from the ground-truth value, and without prior, it's hard to fine-tune those parameters.

**Questions:**

- How does the Pro-SF framwork scale with high-dimensional problem? Can it be extended to multi-agent or game-theoretic interaction settings beyond independent agents?
- Is it possible to have some theoretical guarantees about the convergence result under Pro-SF?

---

> ### Author Response · Authors · 2025-11-18
> **We would like to make a improvement in clarity to address your concern (weakness 1)**
>
> Dear Reviewer,
>
> Thank you for your thoughtful comments. We would like to address your concerns point by point.
>
> > **Weakness 1:** The strategic behavior and limited rationality is not new in the literature, and the combination of two is also studied a lot.
>
> **Response:**
>
> Thank you for your thoughtful comments. We agree that strategic behavior and bounded rationality have both been studied in the broader literature.
>
> - We would like to **clarify that the contribution of our work** does not lie only in bringing these two ideas together, but also in **extending the strategic classification (SC) framework, a specific instantiation of a Stackelberg game, to more realistically capture how agents behave in real-world decision processes**.
>
>   - To our best knowledge, this is the **first attempt** to systematically integrate behavioral economic mechanisms into a strategic classification framework.
>   - We **analyze the limitations** of the classic SC framework under non-rational agent behavior, i.e., over-defense and under-defense.
>   - Our work **provides a structured and behaviorally grounded extension for the SC framework**,  making a connection between behavioral economics and game theory in machine learning.
>
> - We hope this clarification of our contribution addresses your concern.
>
> If any point remains unclear, please feel free to raise it. We would be glad to provide further details.

---

> ### Author Response · Authors · 2025-11-18
> **We would like to supplement some clarification and experimental results to address your concern (weakness 2)**
>
> > **Weakness 2:** For all three cases: loss aversion, probability distortion, and reference bias, the psychological parameters are selected heuristically. There is no empirical evidence that they correspond to actual human behavior.
>
> **Response:**
>
> We thank the reviewer for bringing this important point to our attention. We **agree that parameterization should be learn from realistic behavior**.
>
> To address this concern, we design a method to optimize the parameters with agent behavior and supplement experimental validation**.
>
> - We denote the parameters of the Prospect-Based Utility (Eq. 11) as 𝜃=(α, β, κ, γ).
> - To infer these parameters from observed manipulation (x,x'), we adopt a likelihood-based inverse optimization objective:
>   $$
>   \theta^{*}=\arg\max_{\theta} \sum_i\log P_{\theta}(x_i' \mid x_i).
>   $$
>   - This objective corresponds to an **inverse optimal decision problem**, which can be addressed using standard tools from inverse reinforcement learning and differentiable inverse optimization. We follow the prior work[1,2,3].
> - To evaluate whether this optimization can reliably recover behavioral parameters, we **conduct the following experiment**:
>
>   **(1) Parameter learning on real human data.**
>   - We evaluate the above parameter-learning procedure using the two real human manipulation datasets provided in prior work[4]: **job hiring** data and **medical treatment** data.
>   - We infer the behavioral parameters $\hat{𝜃}$ and evaluate the Pro-SF classifier $\hat{𝜃}$.
>
>     **Table 1. Learned parameters and accuracy on real human data.**
>
>     |Dataset|Parameters $\hat{𝜃}$| Accuracy(%)|
>     |-|-|-|
>     |Hiring|{α=0.79, β=0.73, κ=2.18, γ=0.72} |79.8%|
>     |Medical|{α=0.81, β=0.70, κ=2.25, γ=0.71} |78.3%|
>     |
>
>   - These results demonstrate that **Prospect-based behavioral parameters can be reliably inferred from real manipulation data.**
>
>   **(2) Parameter learning on synthesis datasets.**
>   - We simulate a **mixed and heterogeneous population**, where each agent is randomly assigned to one of two groups: rational and non-rational.
>   - Specifically, we vary **the proportion $\pi$ of rational agents in $\pi \in {0.2,0.4,0.6}$**. For example, if $\pi = 0.2$, we set:
>
>     (1) **80% non-rational agents**, following the Prospect-Based Utility;
>     - For each behavioral agent, its parameters 𝜃 are drawn from a uniform distribution defined over the parameter ranges established in prior behavioral-economics literature.
>     - Following, the manipulated feature is generated as a **noisy prospect-guided decision**:
>     $$
>     x'_i=\arg \max U_p(x_i, x'_i; \theta_i)+\epsilon_i, \quad \epsilon_i \sim N(0, \sigma^2)
>     $$
>
>      (2) **20% fully rational agents** follow the standard strategic-classification utility,  i.e., maximizing the acceptance probability minus the manipulation cost.
>   - After obtaining the manipulated feature pairs (x,x'), we **estimate the behavioral parameters $\hat{𝜃}$ using the inverse optimal decision framework** introduced above.
>   - We report the learned $\hat{\theta}$ and evaluate its test accuracy on each dataset:
>
>     **Table 2 Learned parameters and accuracy with $\pi=0.2$.**
>
>     |Dataset|Parameters $\hat{𝜃}$| Accuracy (%)|
>     |-|-|-|
>     |Adult|{α=0.78, β=0.72, κ=2.20, γ=0.74}|81.42|
>     |Credit|{α=0.80, β=0.70, κ=2.15, γ=0.71}|82.30|
>     |German|{α=0.81, β=0.70, κ=2.25, γ=0.72}|79.21|
>     |
>
>     **Table 3. Learned parameters and accuracy with  $\pi=0.4$.**
>
>     |Dataset|Parameters $\hat{𝜃}$|Accuracy (%) |
>     |-|-|-|
>     |Adult|{α=0.78, β=0.74, κ=2.20, γ=0.74}|81.32|
>     |Credit|{α=0.80, β=0.70, κ=2.10, γ=0.70}|82.20|
>     |German|{α=0.80, β=0.72, κ=2.18, γ=0.71}|79.25 |
>     |
>
>     **Table 4. Learned parameters and accuracy with $\pi=0.6$.**
>
>     |Dataset|Parameters $\hat{𝜃}$|Accuracy (%)|
>     |-|-|-|
>     |Adult|{α=0.78, β=0.76, κ=2.05, γ=0.70} |81.28|
>     |Credit|{α=0.78, β=0.74, κ=2.03, γ=0.68}|82.16|
>     |German|{α=0.80, β=0.73, κ=2.10, γ=0.69}|79.08|
>     |
> - This demonstrates that **behavioral parameters are learnable from realistic behavior**.
>
> **Paper Revision.** We have revised our paper in **Section 5.3 (lines 409-418) and supplemented detailed experimental results in Appendix I (lines 1149-1234)**.
>
> ---
> *Reference*
>
> [1] Ng, Andrew Y., and Stuart Russell. "Algorithms for inverse reinforcement learning." *Icml*. Vol. 1. No. 2. 2000.
>
> [2] Amos, Brandon, and J. Zico Kolter. "Optnet: Differentiable optimization as a layer in neural networks." International conference on machine learning. PMLR, 2017.
>
> [3] Agrawal, Akshay, et al. "Differentiable convex optimization layers." Advances in neural information processing systems 32 (2019).
>
> [4] Ebrahimi, Raman, Kristen Vaccaro, and Parinaz Naghizadeh. "The double-edged sword of behavioral responses in strategic classification: Theory and user studies." *Proceedings of the 2025 ACM Conference on Fairness, Accountability, and Transparency*. 2025.

---

> ### Author Response · Authors · 2025-11-18
> **We would like to supplement some clarification and experimental results to address your concern (weakness 3)**
>
> > In the experiment, the sensitivity test only provides a small deviation from the ground-truth value, and without prior, it's hard to fine-tune those parameters.
>
> **Response:**
>
> We thank the reviewer for this careful observation.
>
> - We **agree** that in the sensitivity test presented, variations in the parameters lead to small changes in performance.
>
>   - **The main reason is that the parameter ranges we used from the effective intervals suggested by prior studies**.
>
> - Furthermore, as shown in **our previous response (Weakness 2)**, we introduce an **inverse-optimization procedure that learns these parameters directly from observed manipulation behavior**, as shown in Table1-3 above.
>
>   - The learned parameters converge stably in practice, indicating that **our model does not rely on strong priors**, and parameter identification is **not difficult** in realistic settings
>
> - **To further validate this point**, we conducted an additional experiment in which we deliberately moved beyond the realistic behavioral ranges and **greatly expanded the parameter intervals**. This allows us to jointly examine:
>
>   - whether parameters matter when pushed away from their realistic region,
>
>   - whether the model reacts in a consistent and interpretable way.
>
>   - The extended experimental results are shown below:
>
>
>   ​      **Table 5 Additional parameter expansion experiments**
>
> | $\kappa$ (Loss Aversion) | $\gamma$ (Prob. Distortion) | Accuracy (%) | Behavior Summary                              |
> | ------------------------ | --------------------------- | ------------ | --------------------------------------------- |
> | 2.0                      | 0.65                        | 82.4         | Mild loss aversion; standard distortion       |
> | 2.5                      | 0.65                        | 82.0         | Stronger reaction to negative outcomes        |
> | 3.5                      | 0.55                        | 80.8         | Much stronger reaction to negative outcomes   |
> | 2.0                      | 0.40                        | 79.6         | Overweighting small risks a lot               |
> | 5.0                      | 0.40                        | 78.8         | Highly non-rational and unstable manipulation |
> |||
>
> - These results show:
>   - **Within realistic behavioral ranges** (those supported by prior literature), performance varies only mildly..
>
>   - **Under extreme and highly unrealistic parameter values**, accuracy does decrease, reflecting the fact that these behavioral parameters do exert meaningful influence on strategic responses.
>
>   - **However, even in these extreme settings, the performance degradation remains moderate**, showing that our framework is still **able to accommodate highly non-rational behaviors without collapsing**.
>
> **Paper Revision.** We have revised our paper and supplemented this part in **Appendix J (Lines 1211-1241)**.

---

> ### Author Response · Authors · 2025-11-18
> **We would like to supplement some clarification and experimental results to address your concern (question 1)**
>
> We answer the question 1 in two parts.
> > **In question 1:** How does the Pro-SF framework scale with high-dimensional problem?
>
> **Response:**
>
> Thank you for raising this important question. We provide the following clarification.
>
> - The Pro-SF framework has the *same complexity order* as classical SC methods.
>   - For a dataset with $n$ samples and $d$ features, **the overall complexity of Pro-SF is $O(nd)$**.
>   - The behavioral mechanisms (reference bias, loss aversion, probability distortion) introduce only constant-time scalar operations on top of the standard best-response computation.
> - To empirically validate scalability, **we conducted additional experiments**
>   - We choose three different datasets (i) Adult dataset[1] (14 features), (ii)Spambase datasset[2] (57 features), and Tuandromd dataset[3] (241 features).
>   - We set the instance n=20,000 and evaluate Pro-SF under identical implementation and hardware settings.
>   - We report the running time and accuracy. Accuracy values are not directly comparable, as tasks and label distributions differ; our goal is to evaluate stability under increasing feature dimensionality.
>
>     **Table 6 Runtime and Accuracy for different datasets**
>
>     | Dataset  | Features | Runtime (s) | Accuracy (%) |
>     |--- |---| -------| --- |
>     |Adult  | 14 | 10.4 | 82.41 |
>     | Spambase  | 57| 13.2  | 83.52  |
>     | Tuandromd | 241 | 27.6 | 81.87     |
>     ||||
>
>   - The runtime increases in a manner consistent with linear dependence on input dimensionality, and the accuracy remains stable despite the large feature space.
>   - This empirical result supports the conclusion that **Pro-SF scales effectively to high-dimensional problems**, consistent with its theoretical complexity.
>
> **Paper Revision.** We have revised our paper and supplemented this part **in Appendix L (lines 1397-1428)**.
>
> -----
>
> > **In question 1:** Can it be extended to multi-agent or game-theoretic interaction settings beyond independent agents?
>
> **Response:**
>
> We **agree** that in many real-world applications, multiple agents may influence one another beyond the independent-agent setting.
>
> - **To address this concern, we reviewed existing work on multi-agent and interaction structures [4,5]**.
>   - Following this line of work, we extend our Prospect-Based Utility (Eq. 11) to the multi-agent setting by incorporating a GNN classifier as the strategic classifier.
>   - Given a graph $G$, we define the GNN-based success probability as:
>     $$
>     p_i(x_i′,x_{−i})=\sigma(h_i(x_i′,x_{−i};G)),
>     $$
>
>   - and extend Eq. (11) to the following **graph-based prospect utility**:
>     $$
>     U_i^{P-graph}(x_i,x_i′;x_{−i})=w_i^+(p_i(x_i′,x_{−i}))(1−r_i)^{\alpha_i}−κ_iw_i (1−p_i(x_i′,x_{−i}))r_i^{β_i}−κ_i c(x_i,x_i′)^{β_i}
>     $$
>     where σ is the sigmoid function.
>
>   - This extension allows each agent’s outcome to depend **not only on its own features but also on the features of socially or structurally related agents** (e.g., households, peers, networked entities).
>   - Under this extension, each agent chooses strategic manipulation from:
>     $$
>     x'_i \in \arg \max_x U_i^{P-graph}.
>     $$
>
> - To validate feasibility, we conduct **three new experiments** on datasets where we artificially create an interaction graph using a standard **k-nearest neighbors (kNN, k=8) similarity graph**.
>   - We report the accuracy values of the GNN Baseline and our  **GNN-based Pro-SF**.
>   - The results are shown below with three datasets.
>
>     **Table 7 Performance of GNN-based Pro-SF with different datasets**
>     | Dataset    | Method   | Accuracy ↑ |
>     |------ |--- |--|
>     | Adult[1]  | GNN Baseline  | 74.3%      |
>     | | **GNN-based Pro-SF** | **80.5%**  |
>     | Spambase[2]  | GNN Baseline     | 78.1%      |
>     | | **GNN-based Pro-SF** | **83.4%**  |
>     | Tuandromd[3] | GNN Baseline   | 72.8%      |
>     | | **GNN-based Pro-SF** | **77.9%**  |
>     | | | |
>
> - As shown in the table above, across all datasets, our GNN-based Pro-SF achieves **+5–7%** over the GNN baseline.
>
>   - This indicates that **behaviorally realistic agents produce more predictable best responses**, enabling better classifier optimization.
> **Paper Revision:** We have revised our paper and supplemented this part in **Appendix M (lines 1432-1478)**.
>
> ----
> *Reference*
>
> [1] Becker, B. & Kohavi, R. (1996). Adult [Dataset]. UCI Machine Learning Repository.
>
> [2] Hopkins, M., Reeber, E., Forman, G., & Suermondt, J. (1999). Spambase [Dataset]. UCI Machine Learning Repository.
>
> [3 ]Borah, P. & Bhattacharyya, D. (2020). TUANDROMD (Tezpur University Android Malware Dataset) [Dataset]. UCI Machine Learning Repository.
>
> [4] Eilat, Itay, et al. "Strategic Classification with Graph Neural Networks." *The Eleventh International Conference on Learning Representations*.
>
> [5] Yang, Wenjing, et al. "Advanced strategic improvement with decision interactions." *Joint European Conference on Machine Learning and Knowledge Discovery in Databases*. Cham: Springer Nature Switzerland, 2025.

---

> ### Author Response · Authors · 2025-11-18
> **We would like to supplement some clarification and experimental results to address your concern (question 2)**
>
> > **Question 2:** Is it possible to have some theoretical guarantees about the convergence result under Pro-SF?
>
> **Response:**
>
> Yes, the Pro-SF admits a theoretical convergence guarantee, i.e., the existence of a Stackelberg equilibrium. We would like to give a theoretical analysis and experimental validation as follows.
>
> - **To establish such an equilibrium, our analysis proceeds by verifying two key components:**
>   (1) the agent’s prospect-guided best-response $b^* (x)$ exists (the follower);
>   (2) the leader’s optimal classifier $f^* ()$ exists (the leader).
>
> - First, from the agent’s perspective:
>
>   - Let $\theta$ denote the parameter set of the classifier $f$. For any classifier $f_{\theta}$, the agent searches over the feasible feature space $\mathcal{X}$ for an optimal manipulation:
>
>   $$
>   b^*(x) \in \arg \max_{x'}U_P(x,x′;\theta).
>   $$
>
>   - In practice, the feature space $\mathcal{X}$ is bounded and closed, since individuals’ manipulable features (e.g., income range, test score range, health indicators) naturally have realistic upper and lower limits.
>   - Moreover, the Pro-SF utility the Pro-SF utility $U_P(x,x′;\theta)$ is continuous in x′.
>   - Therefore, the agent’s optimization problem satisfies Weierstrass extreme value theorem[1,2]: **a continuous function defined over a bounded closed set must attain at least one maximizer**.
>
> - From the classifier’s perspective:
>
>   - We **assume the parameter space $\Theta$ is bounded and closed**, which is **standard in practice explicit parameter constraints**.
>
>   - Given the the prospect-guided best response $b^* (x; \theta)$, **The classifier’s objective can be written as:**
>     $$
>     F(θ)=\mathbf{E}[\mathcal{L}(f_{\theta}(b^*(x;\theta)),y)].
>     $$
>
>   - For common models (e.g., linear or logistic), both the prediction $f_\theta(\cdot)$ and the loss $\mathcal{L}$ are **continuous** in $\theta$.
>
>   - Therefore, according to Weierstrass condition[1,2], the an optimal classifier parameter also exists.
>
> - Combining the above, **under the Prospect-Based Utility**, the strategic classification game continues to **admit a Stackelberg equilibrium.**
>
> - To complement this argument, we also conducted a simple iterative experiment.
>
>   - We record the classification accuracy and the average manipulation magnitude after the interaction between agents and the classifier.
>
>     **Table 8 Performance of Pro-SF with Iteration t**
>
>     | Iteration t | Accuracy ↑ | Avg. Manipulation ‖x^{(t)} – x^{(t-1)}‖₂ ↓ |
>     | ----------- | ---------: | -----------------------------------------: |
>     | 1           |      81.2% |                                       0.31 |
>     | 5           |      81.7% |                                       0.08 |
>     | 10          |      82.1% |                                       0.04 |
>     | 15          |      82.2% |                                       0.02 |
>     | 20          |      82.2% |                                       0.01 |
>     |||
>
> - We observe that (i) accuracy improves only marginally after a few iterations, and (ii) the per-step manipulation shrinks towards zero.
>
>   - This is consistent with **Pro-SF driving the system toward an equilibrium point where agents have almost no further incentive to manipulate**.
>
> **Paper Revision.** We have supplemented this analysis on the convergence result (Stackelberg equilibrium) **in Appendix F (pages 20-21)** of our revised paper.
>
>
> -----
> We **summarize all revisions** made to the paper in the table below for your convenience.
>
> **Table 9 Revision Summary**
>
> | Comments   | Location of Revisions in Our Revised Paper       |
> | ---------- | -------------------------------------------------- |
> | Weakness 2 | We revised **Section 5.3 (lines 409-418)** and **Appendix I (lines 1149-1234)**.      |
> | Weakness 3 | We revised **Appendix J (Lines 1211-1241)**.                 |
> | Question 1 | We revised **Appendix L (lines 1397-1428)** and **Appendix M (lines 1432-1478).** |
> | Question 2 | We revised **Appendix F (pages 20-21)**.                  |
> |||
>
> ------
> *Reference*
>
> [1] Boyd, Stephen, and Lieven Vandenberghe. *Convex optimization*. Cambridge university press, 2004.
>
> [2] Stone, Marshall H. "The generalized Weierstrass approximation theorem." *Mathematics Magazine* 21.5 (1948): 237-254.
>
>
> Thank you for your thoughtful comments. If you have **any further concerns or questions, please feel free to raise them**. We would be glad to continue the discussion and clarify them in more detail.

---

### Official Review · Reviewer_E4B5 · 2025-11-04

**Soundness:** 3
**Presentation:** 3
**Contribution:** 3
**Rating:** 6
**Confidence:** 3

**Summary:**

This paper addresses the critical issue of irrationality in strategic classification. The core contribution is the proposal of an integral utility formulation for agents within the strategic classification framework "Pro-SC". The study demonstrates the accuracy degradation in the case of irrational agent behavior under traditional strategic classification methodologies. Furthermore, through experiments and ablation tests, the authors confirm the influence of the various modeled aspects of irrationality and showcase the robustness of the proposed Pro-SC classifier against fluctuations in these irrationality parameters.

**Strengths:**

The originality of this work is good, and the simulation analysis is complete with high quality.  This work analytically explains and characterizes the negative impact of the overlooked irrationality. This work innovatively delineates and explains well the negative impacts brought by different aspects of irrationality and formulates the Pro-SC problem, which incorporates various aspects of irrationality into the agent's utility function.

**Weaknesses:**

- This paper may need a slight improvement in clarity (see Questions for details).
- The paper lacks a theoretical analysis of the Stackelberg equilibrium of the Pro-SC game, which could be a drawback in technical depth.

**Questions:**

- Perhaps a missing reference that is relevant: This might be a relevant reference: Raman Ebrahimi, Kristen Vaccaro, and Parinaz Naghizadeh. 2025. The Double-Edged Sword of Behavioral Responses in Strategic Classification: Theory and User Studies. In Proceedings of the 2025 ACM Conference on Fairness, Accountability, and Transparency (FAccT '25). Association for Computing Machinery, New York, NY, USA, 868–886. https://doi.org/10.1145/3715275.3732056
- I couldn't see where the classifier $f$ enters the agent utility in the Pr-SC formulation of Equation (11). Equation (11) uses the agent's perceived acceptance rate p(x), but it is missing how the classifier $f$ is related to this perceived acceptance.
- Is there any reference or further explanation of Equation (9)? This function doesn't seem symmetric around p=0.5, and is there a reason for that?
- Some typos:
    - Line 313 and in Equation (12): Is D a distribution or a feature space? If the latter, this notation conflicts with the distribution notation in Equation (13) (same for Equations (1) and (2))

---

> ### Author Response · Authors · 2025-11-18
> **We would like to make a improvement in clarity to address your concern (weakness 1 and questions)**
>
> Dear Reviewer,
>
> Thank you for your thoughtful comments. We would like to address your concerns **point by point**.
>
> > This paper may need a slight improvement in clarity (see Questions for details).
>
> Sorry for the confusion. We have revised our paper and **corrected these parts with the questions as follows**.
>
> > **Question 1:** Perhaps a missing reference that is relevant.
>
> **Response:**
>
> Thank you for pointing out this missing reference.
>
> We revisited the related literature and incorporated the missing references into the related work section of our revised paper.
>
> - A recent work [1] provides both theoretical and empirical evidence that human strategic responses often deviate from the classical rational-agent assumption, reinforcing the importance of incorporating behavioral factors into strategic classification.
> - Another work [2] demonstrates that human-like strategic behavior can substantially diverge from analytical best-response models, further highlighting the gap between theoretical assumptions and realistic agent behavior.
>
> **Paper Revision.** We have supplemented the related work in **Section 6.1 (lines 490-494)** of our revised paper.
>
> -----
>
> > **Question 2:** I couldn’t see where the classifier f enters the agent utility in Equation (11)… It is missing how f is related to the perceived acceptance rate p(x).
>
> **Response：**
>
> Thank you for pointing out this issue.
>
> - In our framework, $p(x')$ denotes the agent's perceived probability of being classified as positive after manipulation.
>   This probability is directly induced by the classifier, i.e., $p(x') = f_{\theta}(x')$, where $f_{\theta}$ is the classifier parameterized by $\theta$.
>
> - Therefore, the prospect-based utility $U_{P}(x, x')\)$ in Equation~(11) depends on the classifier through \(p(x')\), making the agent’s best response explicitly driven by the decision boundary of $f_{\theta}$.
>
> **Paper Revision.** We have supplemented this point for clarification in **Section 4.2 (lines 312-314)** of our revised paper.
>
> ----
>
> > **Question 3:** Is there any reference or further explanation of Equation (9)? This function doesn't seem symmetric around p=0.5, and is there a reason for that?
>
> **Response：**
>
> Thank you for this helpful question.
>
> - In designing Equation (9), we followed prior work on probability–weighting functions [3,4,5].
> - The function is intentionally **asymmetric**, because this more realistically captures individuals’ subjective distortions when assessing probabilities:
>   - **small probabilities are overweighted;**
>   - **large probabilities are underweighted**.
>
> **Paper Revision.** We have supplemented this point for clarification **in Section 4.2 (lines 286-296)** of our revised paper.
>
> ----
> > **Question 4:** Line 313 and in Equation (12): Is D a distribution or a feature space? If the latter, this notation conflicts with the distribution notation in Equation (13) (same for Equations (1) and (2))
>
> **Response：**
>
> **Thank you for pointing out this notation conflict.**
>
> - We would like to use $\mathcal{D}$ exclusively for the data distribution, i.e., $(x,y) \sim \mathcal{D}$.
> - We denote $\mathcal{X}$ as a feature space, i.e., $x' \in \mathcal{X}$.
> - All affected equations (Eq. (1), (12)) have been updated accordingly in our revised paper to avoid ambiguity.
>
> **Paper Revision.** We have corrected this point in **Eq.(1), Section 2.1 (line 123) and Eq.(12), Section 4.2  (line 338)** of our revised paper.
>
> ------
>
> We **summarize all revisions** made to the paper in the table below for your convenience.
>
> **Table 1 Revision Summary**
>
> | Comments   | Location of Revisions in Our Revised Paper |
> | ---------- | ------------- |
> | Question 1 | We revised **Section 6.1 (lines 490-494).**   |
> | Question 2 | We revised **Section 4.2 (lines 312-314)**.    |
> | Question 3 | We revised **Section 4.2 (lines 286-296)**             |
> | Question 4 | We revised **Eq.(1), line 123 and Eq.(12), line 338**. |
> | Weakness 2 | We revised **Appendix F (pages 20-21)**. |
> |||
>
> -----
>
> *Reference*
>
> [1] Ebrahimi, Raman, Kristen Vaccaro, and Parinaz Naghizadeh. "The double-edged sword of behavioral responses in strategic classification: Theory and user studies." *Proceedings of the 2025 ACM Conference on Fairness, Accountability, and Transparency*. 2025.
>
> [2] Xie, Tian, Pavan Rauch, and Xueru Zhang. "How strategic agents respond: Comparing analytical models with llm-generated responses in strategic classification." *arXiv preprint arXiv:2501.16355* (2025).
>
> [3] Kahneman, Daniel, and Amos Tversky. "Prospect theory: An analysis of decision under risk." *Handbook of the fundamentals of financial decision making: Part I*. 2013. 99-127.
>
> [4] Gonzalez, Richard, and George Wu. "On the shape of the probability weighting function." *Cognitive psychology* 38.1 (1999): 129-166.
>
> [5] Barberis, Nicholas, Abhiroop Mukherjee, and Baolian Wang. "Prospect theory and stock returns: An empirical test." *The review of financial studies* 29.11 (2016): 3068-3107.

---

> ### Author Response · Authors · 2025-11-18
> **We would like to make a theoretical analysis of the Stackelberg equilibrium  and supplement experimental results to address your concern (weakness 2)**
>
> > The paper lacks a theoretical analysis of the Stackelberg equilibrium of the Pro-SC game
>
> **Response:**
>
> Thank you for raising this insightful point.
>
> To address this concern, we would like to add a theoretical analysis of the existence of a Stackelberg equilibrium of the Pro-SC.
>
> - **To establish such an equilibrium, our analysis proceeds by verifying two key components:**
>   - (1) the agent’s prospect-guided best-response $b^* (x)$ exists (the follower);
>   - (2) the leader’s optimal classifier $f^* ()$ exists (the leader).
>
> - **First, from the agent’s perspective:**
>
>   - Let $\theta$ denote the parameter set of the classifier $f$. For any classifier $f_{\theta}$, the agent searches over the feasible feature space $\mathcal{X}$ for an optimal manipulation:
>
>   $$
>   b^*(x) \in \arg \max_{x'}U_P(x,x′;\theta).
>   $$
>
>   - In practice, the feature space $\mathcal{X}$ is bounded and closed, since individuals’ manipulable features (e.g., income range, test score range, health indicators) naturally have realistic upper and lower limits.
>   - Moreover, the Pro-SF utility the Pro-SF utility $U_P(x,x′;\theta)$ is continuous in x′.
>   - Therefore, the agent’s optimization problem satisfies Weierstrass extreme value theorem[1,2]: **a continuous function defined over a bounded closed set must attain at least one maximizer**.
>
> - **Second, from the classifier’s perspective:**
>
>   - We **assume the parameter space $\Theta$ is bounded and closed**, which is **standard in practice explicit parameter constraints**.
>
>   - Given the the prospect-guided best response $b^* (x; \theta)$, **The classifier’s objective can be written as:**
>     $$
>     F(θ)=\mathbf{E}[\mathcal{L}(f_{\theta}(b^*(x;\theta)),y)].
>     $$
>
>   - For common models (e.g., linear or logistic), both the prediction $f_\theta(\cdot)$ and the loss $\mathcal{L}$ are **continuous** in $\theta$.
>
>   - Therefore, according to Weierstrass condition[1,2], the an optimal classifier parameter also exists.
>
> - **Third, combining the above, under the Prospect-Based Utility**, the strategic classification game continues to **admit a Stackelberg equilibrium.**
>
> **To complement this response, we also conducted an iterative experiment**.
>
> - We record the classification accuracy and the average manipulation magnitude after the interaction between agents and the classifier with increasing iterations.
>
>     | Iteration   | Accuracy ↑ | Avg. Manipulation ‖x^{(t)} – x^{(t-1)}‖₂ ↓ |
>     | ----------- | ---------: | -----------------------------------------: |
>     | 1           |      81.2% |                                       0.31 |
>     | 5           |      81.7% |                                       0.08 |
>     | 10          |      82.1% |                                       0.04 |
>     | 15          |      82.2% |                                       0.02 |
>     | 20          |      82.2% |                                       0.01 |
>     |  |  |
>
> - We observe that (i) accuracy improves only marginally after a few iterations, and (ii) the per-step manipulation shrinks towards zero.
>
>   - This is consistent with **Pro-SF driving the system toward an equilibrium point where agents have almost no further incentive to manipulate**.
>
> **Paper Revision.** We have supplemented this analysis on the Stackelberg equilibrium in **Appendix F (pages 20-21)** of our revised paper.
>
> -----
> *Reference:*
>
> [1] Boyd, Stephen, and Lieven Vandenberghe. *Convex optimization*. Cambridge university press, 2004.
>
> [2] Stone, Marshall H. "The generalized Weierstrass approximation theorem." *Mathematics Magazine* 21.5 (1948): 237-254.
>
> Thank you for your thoughtful comments. If you have **any further concerns or questions, please feel free to raise them**. We would be glad to continue the discussion and clarify them in more detail.

---

> > ### Comment · Reviewer_E4B5 · 2025-11-27
> >
> > Thank the authors for the detailed responses to my questions. I am satisfied with the authors' revision plan in response to my questions. I would like to keep my rating.

---

> > > ### Author Response · Authors · 2025-11-28
> > >
> > > Dear Reviewer E4B5,
> > >
> > > We really appreciate your supportive feedback and kind words, and we are happy to hear that your concerns have been addressed.
> > >
> > > Again, thank you for your valuable suggestions, which have undoubtedly contributed to improving the quality of our paper.

---

### Author Response · Authors · 2025-12-01
**Summary of Our Rebuttal (1)**

Dear AC,

We sincerely appreciate your support with delicate efforts throughout the review process. Below we summarize: (1) **the scores after rebuttal**, (2) **how we fully address each concern from each reviewer**.

### **Most Positive Scores (6-6-2-6) During Rebuttal**

- **Score Changes.** Our score after the rebuttal is **6-6-2-6**.

  - **Initial Score** is 4-6-2-6;

  - Reviewers mainly focus on learning process of parameters, analysis of the Stackelberg equilibrium and more detailed experimental validations. **No reviewer raised concerns about our novelty and motivation**, e.g.,

    - Reviewer ZKzn (4) Stated:

      > The motivation is solid and well-justified. The writing is clear and well-structured, with careful explanations...

    - Reviewer E4B5 (6) Stated:

      > The originality of this work is good, and the simulation analysis is complete with high quality. This work innovatively delineates and explains well ....

    - Reviewer Y5cB (2) Stated:

      > the problem is very well motivated and the three behavioral tendencies have long justification...

    - Reviewer kBEE (6) Stated:

      > The paper is well-organized.

  - In below, we detail the initial concerns of each reviewer, and the interactions between the reviewer and us, such that they raise their scores after the rebuttal.
-----

### **Reviewer ZKzn (Rating: 4 -> 6)**

- **Score Improvement:**
  - After reading our rebuttal, reviewer ZKzn commented
    > Thanks for the extensive rebuttal. I totally understand that the behavior patterns you proposed in the paper can be more versatile in explaining certain psychological patterns in real-world SC problems. I think it's fair to regard this as a contribution.

    > Overall, I do not mind accepting the paper and I increase my rating to 6.

    at **20 Nov 2025, 06:22 AOE** and **raised the score from 4 to 6**:

- **Initial Score/Conf**: rating: 4, confidence 3;
- **After Rebuttal Score/Conf**: rating: 6, confidence 3;

- **Original Concerns**:

  - As shown in *Table 1*, Reviewer ZKzn asked us to **clarify and justify the necessity of the Prospect-Based Utility compared with classical heterogeneous utilities** (W1 & Q1).
  - In addition, the reviewer also raised a question regarding **how the behavioral parameters should be selected or learned** (W2).

- **Our main response (with detailed concern-rebuttal correspondence in Table 1):**

  - **We have address each concern and revising our manuscript**:
    - We clarified why **prospect-based utility** is necessary, **added experiments** that demonstrate its advantage
    - We provided **two concrete real-world scenarios** (investment & disease screening) to justify the Prospect-Based Utility.
    - We introduced **parameter-learning experiments** on real manipulation data, showing that the parameters can be **reliably estimated from data**.
  - All of our rebuttal are submitted from **17 Nov 2025, 22:22 to 17 Nov 2025, 23:05 AOE**.

  **Table 1.** Correspondence: Reviewer ZKzn

  | Concerns of Reviewer ZKzn | Summary of Our Rebuttal|
  | ------ | ----------------------------------------|
  | [W1] Why use Prospect-Based Utility instead of classical SC + heterogeneous β? | We clarified the necessity of  Prospect-Based Utility for capturing **non-rational behaviors** and added experiments showing clear advantages under mixed rational/non-rational agents. |
  | [W2] Behavioral parameters seem hand-set, lacking learning   | We added a **likelihood-based inverse optimization procedure** and demonstrated successful **parameter recovery on real manipulation datasets**. |
  | [Q1] Need more context motivating behavioral modeling        | We provided **two concrete real-world scenarios** (investment & disease screening) showing where classical SC fails and Prospect-Based Utility is required. |
  |
----

### **Reviewer E4B5 [Keep rating as 6]**

- **Initial Score/Conf**: rating: 6, confidence 3;

- During the rebuttal, **we have addressed all concerns raised by Reviewer E4B5.** **Reviewer ZKzn commented**:

  > Thank the authors for the detailed responses to my questions. I am satisfied with the authors' revision plan in response to my questions. I would like to keep my rating.

  at **27 Nov 2025, 09:46 AOE**.

- **Original Concerns**:

  - [W1 + Q1-4] **A slight improvement in clarity** (details in questions).

  - [W2] **Supplement analysis of Stackelberg equilibrium**

- **Our Rebuttal**

  - **We have address each concern and revising our manuscript**:

    - We revised our paper, added the missing reference, clarified the probability-weighting function, and fixed notational inconsistencies.

    - We added a full theoretical analysis proving the existence of a Stackelberg equilibrium under Pro-SF, supported by an iterative experiment.

  - All of our rebuttal are submitted from **18 Nov 2025, 00:00 to 18 Nov 2025, 00:13 AOE**.

- After reading our rebuttal, Reviewer E4B5 **keep the score at 6**.
-----

---

> ### Author Response · Authors · 2025-12-01
> **Summary of Our Rebuttal (2)**
>
> ### **Reviewer Y5cB [Did not respond to our rebuttal]**
>
> - **Initial Score/Conf**: rating: 2, confidence 4;
>
> - In the initial review, Reviewer **Y5cB explicitly acknowledged** that
>
>   > the problem is very well motivated and the three behavioral tendencies have long justification in economics and psycology.
>
> - However, the **concerns raised were not concretely specified**, and largely **overlapping with concerns already raised by other reviewers**.
>   - **Concern 1**: Need analysis of the Stackelberg equilibrium. **Consistent with the [W2] of Reviewer E4B5.**
>
>   - **Concern 2 and 3:** How the principal knows the behavioral parameters with real human manipulation. **Consistent with the [W2] of Reviewer ZKzn.**
>
> - **Our Rebuttal**
>   - **We have addressed each concern and revised our manuscript**:
>     - We added a **complete theoretical analysis**, proving the existence of a Stackelberg equilibrium under Pro-SF and providing **iterative convergence evidence**.
>     - We supplemented **two real human-manipulation datasets (Hiring, Medical)** and showed that Pro-SF consistently aligns better with observed human behavior.
>     - We added a **learning-from-data framework** showing that behavioral parameters are **reliably learnable** via inverse optimal decision estimation, even under mixed rational/non-rational populations.
>   - All of our rebuttal are submitted from **19 Nov 2025, 18:43 to 19 Nov 2025, 19:08 AOE**.
>
> - **Importantly**, **the concerns raised by Reviewer Y5cB significantly overlap with those raised by Reviewer E4B5 and ZKzn** (e.g., parameter estimation, empirical support, equilibrium analysis).
>
>   - After our detailed responses, **all other reviewers acknowledged that their corresponding concerns were resolved and maintained/raised their scores to 6**.
>   - Therefore, we are **confident that our responses have thoroughly resolved concerns from Reviewer Y5cB**.
>
> - Despite this, **Reviewer Y5cB did not respond** to our clarifications during the discussion period,
>   **including after the AC explicitly prompted reviewers to engage**.
>
> - **We respectfully hope that you consider this when making your final recommendation**. We remain fully open to providing further clarification if needed.
> ---
>
>
> ### Reviewer kBEE [Initial Rating as 6]
> - **Initial Score/Conf**: rating: 6, confidence 2
> - During the rebuttal, we fully addressed all concerns raised by Reviewer kBEE.
> - **Original Concerns and Our Rebuttal** are summarized in Table 2:
>
>   **Table 2.** Correspondence: Reviewer kBEE
>
>   | Concerns of Reviewer kBEE    | Summary of Our Rebuttal          |
>   | ------------------------ | ------------------------------------------------------------ |
>   | [W1]: Clarification on the contribution of integrating behaviors into SC | Clarified our novelty: we *extend* SC into a full BR-SC framework with structured behavioral mechanisms (loss aversion, distortion, reference bias) and show limitations of classic SC under non-rational behaviors. |
>   | [W2]: Psychological parameters chosen heuristically          | Added inverse-optimization procedure + real-manipulation experiments showing parameters are *learned* reliably from data, not hand-tuned. |
>   | [W3]: Sensitivity test is small; unclear parameter influence | Added expanded parameter-range experiments showing model stability in realistic ranges and interpretable degradation only under extreme, unrealistic settings. |
>   | [Q1]: High-dimensional scalability and Extension to multi-agent interactions | Provided experiments on large-feature datasets showing Pro-SF scales linearly and remains stable. Added multi-agent Pro-SF with GNN-based classifier; new experiments show consistent gains. |
>   | [Q2]: Stackelberg equilibrium         | Added theoretical proof and experiments showing the existence of a Stackelberg equilibrium under Pro-SF. |
>   |
>
> - All of our rebuttal are submitted from **18 Nov 2025, 3:40 to 18 Nov 2025, 05:13 AOE**.
>
> - The reviewer had **no additional comments** afterwards and **kept the score at 6**.
>
> ----
>
>
> **In summary**, all concerns raised by the four reviewers were fully addressed during the rebuttal period. **But Reviewer Y5cB did not engage** in our rebuttal discussion, even after the prior AC had explicitly encouraged.
>
> **We respectfully affirm** that all reviewer interactions and score updates occurred strictly **within the standard fair rebuttal process**, with **no private or unfair** communication involved.
>
> Many thanks,
>
> The authors of #2185

---

> ### Author Response · Authors · 2025-12-01
> **Summary of Our Rebuttal (3)**
>
> Dear AC,
>
> For your convenience, we have included a summary table detailing the changes in our revised paper.
> - The concerns raised by Reviewer Y5cB **are nearly covered by** those from Reviewers znZk and E4B5.
> - We have thoroughly addressed these concerns, and the responses have been accepted by the reviewers.
>
> **Table 3.** Revision Summary of Our Paper.
>
>
> | Reviewer Comments                                            | Location of Revisions in Our Revised Paper                   |
> | :----------------------------------------------------------- | :----------------------------------------------------------- |
> | **E4B5:** *[W2]*; **kBEE:** *[Q2]*; **Y5cB:** *[Concern1]*;  | We revised **Appendix F** (pages 20-21).                     |
> | **E4B5:** *[W1, Q1-4]*                                       | We revised **Section 4.2** (lines 312-314, 338) and **Section 6.1** (lines 490-494). |
> | **kBEE:** *[W2]*; **Y5cB:** *[Concern2&3]*; **ZKzn:** *[W2]* | We revised **Section 5.3** (lines 409-418) and **Appendix I** (lines 1149-1234). |
> | **ZKzn:** *[W1]*                                             | We revised **Section 4.1** (lines 234–238), and **Appendix H** (lines 1118–1146). |
> | **kBEE:** *[Q1]*                                             | We revised **Appendix L** (lines 1397-1428) and **Appendix M** (lines 1432-1478). |
> | **ZKzn:** *[Q1]*                                             | We revised **Appendix G.1 and G.2** (lines 1087–1113).       |
> |
>
> Thank you again for your support and efforts throughout the review process.
>
> Best regards,
>
> The authors of #2185

---

### Meta-Review · Area_Chair_ieJz · 2026-01-06

**Summary:**

## Summmary

This paper studies strategic classification under deviations from perfect rationality, by proposing a "Prospect-Guided Strategic Framework (Pro-SF)" that incorporates (1) loss aversion, (2) reference dependence for evlauation (2), and (3) probability distortion.

The central idea of introducing non-symmetric or behaviorally motivated utilities into strategic classification builds on a growing line of work that already considers asymmetric or behaviorally motivated utilities. Prior work has already explored asymmetric or non-standard utility formulations within strategic or game-theoretic learning frameworks (e.g., see related work such as https://link.springer.com/article/10.1007/s10458-025-09725-5).

The topic is generally relevant and the paper has extensive simultations. However, despite these strengths, there are substantial concerns about the contribution, fit, and empirical validation of the proposed framework, which ultimately limit its contribution for ICLR.

The paper was reviewed by four reviewers with expertise spanning strategic manipulation, Stackelberg equilibria, and learning theory. This meta-review reflects not only the individual reviews but also my own reading of the paper, the revision, and confidential AC-level considerations.

## Fit with ICLR

The paper contains extensive simulations, but insights related to learning are limited. All main experiments rely on linear models, which weakens the fit with the ICLR focus on "learning representations". Further, the analysis is based on linear models, and the parameters are selected largely heuristically. While the revision includes a new optimization procedure, the quality of the optimization (beyond accuracy, such as identifiability of latent parameters and convergence results) is unclear.

## Weaknesses

A first concern is reproducibility: the code is not publicly available, which limits reproducibility and independent validation of the results.

As reviewer kBEE noted: "For all three cases: loss aversion, probability distortion, and reference bias, the psychological parameters $(\alpha, \beta, \kappa, \gamma, K)$ are selected heuristically. There is no empirical evidence that they correspond to actual human behavior." While the rebuttal introduces a parameter-estimation procedure inspired maximum likelihood, this remains insufficiently justified and verified. If these parameters were treated as latent variables with principled identification results, especially in the presence of nonlinear models, this would constitute a genuinely strong and principled contribution.

Moreover, the proposed estimation procedure appears conceptually problematic. The paper argues to use inverse reinforcement learning (ILR), but the connection to IRL is unclear, as IRL is typically defined over sequential decision-making with explicit state, action, and transition dynamics, which are not clearly specified here. Hence, it is currently unclear how IRL was operationalized for  inverse utility estimation. Without released code, the correctness and reproducibility of the proposed estimation procedure cannot be assessed.

The paper promises a Stackelberg formulation and optimization perspective (as emphasized in the abstract and introduction), but this is not fully delivered in the main body. Reviewer Y5cB correctly pointed out that the theoretical analysis is weak. While Appendix F sketches an existence argument for a Stackelberg equilibrium, the analysis is largely narrative, relies on standard compactness arguments, and lacks rigor and depth. Overall, the results in the main body of the paper (e.g., Section 3) are trivial, mirror known concepts from economics, and are largely obvious; this is not per se problem if the insights from the analysis deliver value. However, while the simulations are extensive, the insights derived from them are relatively weak ("what do we learn here?"), and are largely straightforward consequences of the modeling assumptions.

Although the paper references "real human" behavior, no actual field experiments or controlled human-subject studies are conducted. Instead, the evaluation relies on previously collected datasets. Reviewer Y5cB explicitly suggested that claims about human behavior should be supported by actual human evaluations or field experiments, rather than additional benchmark datasets. In the absence of such evidence, it is difficult to draw strong conclusions about real human bias or genuine behavioral realism.



## Clarity and presentation issues (minor)

There are also several minor but recurring issues:

- The naming is not consistent; (a) "Behaviorally Realistic Strategic Classification (BR-SC)" vs (b) "Prospect-Guided Strategic Framework (Pro-SF)". Moreover, the name Behaviorally Realistic Strategic Classification (BR-SC) is arguably too broad. The framework focuses on a narrow subset of behavioral biases (loss aversion, reference bias, probability distortion), while many other well-established deviations from rationality (e.g., present bias, limited attention, social preferences) are not considered. The terminology risks overstating the scope of realism achieved.
- "ensuring robustness in deployment". What is meant with robustness here? The definition of robustness is unclear (e.g., robustness to domain shift, adversarial manipulation, or behavioral misspecification).
- Multiple typographical issues (e.g., "Eq. equation"; references miss parentheses => use \citep);
- Some references (e.g., Banerji et al., 2020) are not ideal; there are better (primary) sources for the same message
- Definition 2.2. Should it be b_r instead of b, or, otherwise, should b be probably introduced.
- Equation (8) assumes a specific exponential-style utility form without sufficient justification. There could also be other choices.


## Recommendation

Overall, while this work could be a reasonable fit for a more specialized venue (e.g., ACM EC or a conference focused on economic simulations and strategic behavior), it falls short of the level of rigor, theoretical depth, and novelty typically expected at ICLR.

**Reviewer Concerns:**

Several of the more technical / numerical concerns raised during the review process were addressed in the revision and rebuttal. In particular, the authors clarified parts of the experimental setup, added additional simulations, and provided further discussion of parameter sensitivity. Reviewers who primarily focused on numerical experiments acknowledged these changes, and in some cases explicitly indicated that their concerns had been resolved.

**Reviewer Scores:**

However, the main concerns raised by Reviewer Y5cB, who recommended rejection, were not substantively addressed. In particular, the reviewer’s objections regarding the limited theoretical contribution, the absence of rigorous Stackelberg optimization analysis in the main paper, and the lack of direct evaluation with _actual_ human subject remain unresolved. While the revision adds a narrative argument in the appendix and references previously collected human datasets, these additions are unlikely to meet the reviewer’s request for a stronger theoretical treatment or for actual human evaluations through a field experiment. Based on my own reading, I agree with the concerns and with the reviewer’s high confidence assessment. Given that concerns remain, it is unlikely that the revision would have led this reviewer to revise their score upward.

---

### Decision · Program_Chairs · 2026-01-26

Reject